# Embracing Discrete Search: A Reasonable Approach to Causal Structure Learning

**Marcel Wienöbst[1], Leonard Henckel[2], Sebastian Weichwald**[*,3]
[1]University of Luebeck, [2]University College Dublin, [3]University of Copenhagen

## Abstract

We present FLOP (Fast Learning of Order and Parents), a score-based causal discovery algorithm for linear models. It pairs fast parent selection with iterative Cholesky-based score updates, cutting run-times over prior algorithms. This makes it feasible to fully embrace discrete search, enabling iterated local search with principled order initialization to find graphs with scores at or close to the global optimum. The resulting structures are highly accurate across benchmarks, with near-perfect recovery in standard settings. This performance calls for revisiting discrete search over graphs as a reasonable approach to causal discovery.

## 1 Introduction

Learning about the directed acyclic graph (DAG) underlying a system's data-generating process from observational data under causal sufficiency is a fundamental causal discovery task (Pearl, 2009). Score-based algorithms address this task by assigning penalized likelihood scores to each DAG and seeking graphs whose scores are optimal. Identifiability theory asks when such score-optimal graphs identify the target graph (or its equivalence class) in the infinite-sample limit, with various results under different assumptions and scores (Chickering, 2002; Nandy et al., 2018).

Exact algorithms, that are guaranteed to find a score-optimal graph, have exponential run-time and are feasible up to roughly 30 variables (Koivisto & Sood, 2004; Silander & Myllymäki, 2006). For larger graphs, local search must be employed, which evaluates neighbouring graphs to find graphs with better scores; canonical moves for this hill climbing are single edge insertions, deletions, or reversals (Heckerman et al., 1995). In the sample limit, greedy discrete search with a neighbourhood notion that respects score equivalence provably finds a graph with optimal score (Chickering, 2002). In finite samples, scores are inexact and local search may get stuck in local optima or, as we demonstrate, even find graphs with better scores than the true graph. Finite-sample performance is a practical challenge, despite the mature identifiability theory and asymptotic guarantees.

Continuous optimization methods have emerged as a popular alternative. For example, NOTEARS encodes acyclicity as a smooth constraint and optimizes a surrogate objective (Zheng et al., 2018), with many follow-ups (Bello et al., 2022; Rolland et al., 2022). Their supposed advantages have been questioned empirically and conceptually (Reisach et al., 2021; 2023; Ng et al., 2024). Further, NP-hardness results often cited to dismiss discrete search do not apply to the commonly considered discovery settings: The standard hardness constructions rely on data-generating processes that involve unobserved variables and cannot be represented by a DAG over only the observed variables (Chickering, 1996; Chickering et al., 2004). When the distribution is representable by a sparse DAG, discrete procedures asymptotically recover the target graph with polynomially many independence tests or score evaluations (Claassen et al., 2013; Chickering & Meek, 2015).

One of the core issues of score-based methods in practice are finite-sample induced local optima (Nielsen et al., 2003). Hence, the best-performing heuristics in benchmarks (Rios et al., 2025) are either able to escape local optima, for example through simulated annealing (Kuipers et al., 2022), or realize larger neighborhoods (Pisinger & Ropke, 2018), such as recent order-based methods (Lam et al., 2022; Andrews et al., 2023) with effective reinsertion moves rather than only swapping neighboring nodes (Teyssier & Koller, 2005; Scanagatta et al., 2015). This helps explain the strong performance of the order-based BOSS algorithm (Andrews et al., 2023) and more recent

---

[*]Correspondence: `sweichwald@math.ku.dk`

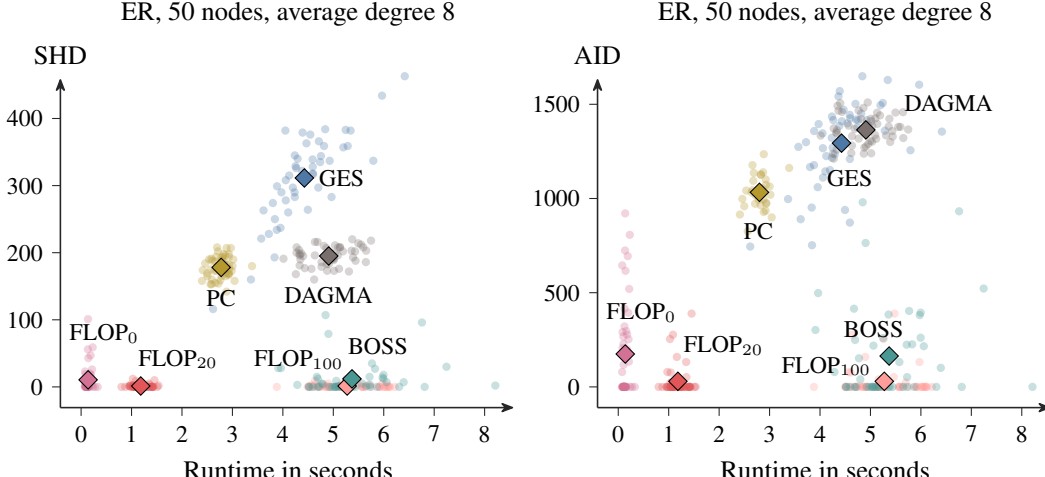

Figure 1: Run-time plotted against Structural Hamming Distance (left) and Ancestor Adjustment Identification Distance (Henckel et al., 2024) (right) between the CPDAGs learned on linear ANM data and the target CPDAG corresponding to the underlying Erdős-Renyi generated DAG with 50 nodes, average degree 8 and 1000 samples drawn. Every point corresponds to one of 50 random instances; diamonds indicate averages. FLOP variants differ in the number of ILS restarts to escape local optima. The fraction of instances with exact CPDAG recovery is 40% for BOSS and $\text{FLOP}_0$ and 60% for $\text{FLOP}_{20}$ and $\text{FLOP}_{100}$, and zero for the remaining algorithms.

order-based local searches (Li et al., 2025) on common causal discovery benchmarks. Continuous relaxations also alter the search space traversal, yet they have not matched this performance and introduce additional challenges, for example optimization complexity, convergence issues, edge thresholding, and having to resort to surrogate objectives.

**Contributions.** We introduce FLOP (Fast Learning of Order and Parents), a score-based structure learning algorithm for linear additive noise models that fully embraces discrete search, and offer a Rust implementation at github.com/CausalDisco/flopsearch ready-to-use from Python via `pip install flopsearch`. The FLOP algorithm adopts reinsertion- and order-based exploration of DAGs (Andrews et al., 2023) and adds four components that enable aggressive search for graphs with optimal BIC score. First, we simplify parent selection by re-initializing from the parent sets learned for the previous order, which reduces compute and memory cost without degrading performance (Section 3.1). Second, we accelerate score computations for the linear Gaussian BIC via efficient iterative updates of Cholesky factorizations, which amortize cost across local moves (Section 3.2). Third, we develop a principled initialization that, compared with a random initial order, reduces local parent selection failures on ancestor-descendant pairs that are far apart and only weakly dependent (Section 4.1). Fourth, the computational gains allow us to employ an iterated local search (ILS) metaheuristic to escape local optima (Section 4.2). On standard benchmarks, the order-based methods BOSS and FLOP achieve strong accuracy at favorable run-time (see Figure 1). FLOP's run-time advantage can be translated into higher accuracy by extending the ILS budget, with $\text{FLOP}_k$ denoting $k$ iterations. By treating compute budget as a hyperparameter, our work highlights the link between run-time and finite-sample accuracy in causal discovery.

## 2 PRELIMINARIES

We consider the problem of learning about the acyclic graph structure underlying linear additive noise models (ANMs) from observational data. For a causal DAG $G = (V, E)$ with node set $V = \{1, \ldots, p\}$ and edge set $E \subsetneq \{i \to j \mid i, j \in V\}$, a linear ANM is defined by a weight matrix $W \in \mathbb{R}^{p \times p}$ with $W_{i,j} \neq 0 \iff i \to j \in E$ and a vector $N = [N_1, \ldots, N_p]^T$ of jointly independent, real-valued, zero-mean noise variables with finite fourth moment; the observed variables $X = [X_1, \ldots, X_p]^T$ are then defined by $X = W^T X + N$.

---

**Algorithm 1:** Reinsertion-based local search as proposed by Andrews et al. (2023).

---

**input** : Data set $D$ over $p$ variables.
**output:** A CPDAG $G$.

1   $\tau :=$ initial order of $\{1, \ldots, p\}$ `// Below, `$\tau_i$` refers to the `$i$`-th element of `$\tau$
    `/* `$P_v$`, `$\ell_v$` contain the parents and the local score of node `$v$` */`
2   **foreach** $i \in \{1, \ldots, p\}$ **do** $(P_{\tau_i}, \ell_{\tau_i}) :=$ growShrink $(\tau_i, \tau_{1:i-1}, D)$
3   **repeat**
4      $(\tau^{\text{old}}, \ell^{\text{old}}) := (\tau, \ell)$
5      **for** $v \in \tau^{old}$ **do**
6         $(\tau, P, \ell) :=$ reinsert $(\tau, P, \ell, v, D)$ `// Optimal reinsertion for `$v$`.`
7      **end**
8   **until** sum $(\ell) \geq$ sum $(\ell_{old})$
9   **return** CPDAG of the DAG defined by parent sets $P_1, \ldots, P_p$

---

We denote the parents of $v \in V$, that is all $u \in V$ such that $u \to v \in E$, by Pa$(v)$. Every DAG $G = (V, E)$ can be associated to at least one linear order $\tau$, also called *topological* order, of the nodes such that $u \to v \in E$ implies $u$ coming before $v$ in $\tau$. A DAG is called Markovian to a probability distribution if every variable is independent of its non-descendants (all nodes not reachable from it with a directed path) given its parents. We denote conditional independence by $\perp$.

As score to optimize, we choose the Bayesian Information Criterion (Schwarz, 1978), BIC for short, which is the common choice in score-based structure learning for ANMs and is asymptotically consistent under the faithfulness assumption (Koller & Friedman, 2009), which we employ throughout this paper. For a DAG $G = (V, E)$ and a data set $D$ containing $n$ observations, it is defined as $k \cdot \ln(n) - 2 \ln(\hat{L})$ with $k$ being the number of parameters and $\hat{L}$ the maximized likelihood for the given DAG and data. The score can be decomposed into local scores $\text{BIC}_D(G, X) = \sum_{v \in V} \ell_D(X_v, X_{\text{Pa}(v)})$ and, for linear models with Gaussian noise, each local score is given by $\ell_D(X_v, X_{\text{Pa}(v)}) = n \log(\widehat{\text{Var}}_D(X_v \mid X_{\text{Pa}(v)})) + \lambda \ln(n) |\text{Pa}(v)|$ with $\lambda$ being a penalty parameter.[1] The BIC is score-equivalent: Its value is the same for Markov equivalent DAGs (Verma & Pearl, 1990) that imply the same conditional independencies. In fact, with observational data under causal sufficiency and without additional assumptions, only such an equivalence class of DAGs is identifiable. Throughout, our target object is therefore the equivalence class of the underlying DAG, represented by a completed partially directed acyclic graph (Andersson et al., 1997). Methods that internally optimize over DAGs, such as continuous relaxations, or rely on extra assumptions to identify a unique DAG, such as non-Gaussian linear models (Shimizu et al., 2011), or noise-variance conditions (Park, 2020), need to be evaluated via the corresponding CPDAG for a fair comparison; evaluating a single DAG as if identified is arbitrary under our assumptions and can be misleading.

We employ principles from the BOSS algorithm (Andrews et al., 2023) to optimize the BIC score over DAGs: We traverse the space of topological orders of DAGs, iteratively moving to orders that result in better scoring DAGs when selecting parents accordingly. Candidate orders are generated by taking a variable and reinserting it at another position. Given an order $\tau$, we use the grow-shrink procedure (Margaritis, 2003) to construct a parent set for each variable $v$ from its prefix, the variables preceding it in $\tau$, and score the resulting DAG. Algorithm 1 shows the BOSS reinsertion strategy, which we build on for the improved search in FLOP.

## 3   Scaling Up Order-Based Search

This section presents two speedups for order-based local search with reinsertion moves, which yield significantly faster run-times than the grow-shrink trees used in BOSS. FLOP still uses grow-shrink to obtain DAGs from orders, but in a way that exploits the local, iterative search moves and scoring.

---

[1]Here, one needs to assume that the empirical covariance matrix is non-degenerate and numerically well-conditioned, excluding cases such as zero-variance noise, high-dimensional settings with $n < p$, or variance blow-up severe enough to cause numerical instability.

---

**Algorithm 2:** Find the best-scoring reinsertion of node $v$ in order $\tau$ given data $D$.

```
1  function reinsert(τ, P, ℓ, v, D) // P stores parents, ℓ local scores.
2  │    i := position of v in τ
3  │    (τ̂, P̂, ℓ̂) := (τ, P, ℓ)
4  │    foreach j ∈ {i + 1, ..., p} do // Test reinsertions at later positions.
5  │    │    (τⱼ₋₁, τⱼ) := (τⱼ, τⱼ₋₁)        // Swap τⱼ₋₁ one position to the right.
   │    │                                 // Compute parents for changed prefixes of τⱼ₋₁ and τⱼ.
6  │    │    (ℓ_τⱼ, P_τⱼ) := growShrink(τⱼ, τ₁:ⱼ₋₁, D, P_τⱼ, l_τⱼ, +τⱼ₋₁)
7  │    │    (ℓ_τⱼ₋₁, P_τⱼ₋₁) := growShrink(τⱼ₋₁, τ₁:ⱼ₋₂, D, P_τⱼ₋₁, l_τⱼ₋₁, −τⱼ)
8  │    │    if sum(ℓ) < sum(ℓ̂) then (τ̂, P̂, ℓ̂) := (τ, P, ℓ)
9  │    end
10 │    foreach j ∈ {i − 1, ..., 1} do ... // Analogous for earlier positions.
11 end
12 return τ̂, P̂, ℓ̂
```

---

### 3.1 STARTING GROW-SHRINK FROM THE PREVIOUS PARENT SET

During the scoring of node-reinsertions, each node's candidate parent set, that is, the nodes coming before it in the order, changes by at most one node being inserted or deleted from its prefix. Consider Algorithm 2 that finds the best reinsertion for node $v$ currently at position $i$ in order $\tau$. The possible reinsertions of $v$ can be efficiently evaluated by performing a sweep from position $i$ to the right (and also to the left; analogous code omitted), moving it to position $i + 1$, $i + 2$, and so on, by swapping it rightward. At each step, the prefix of node $v$ increases by exactly one element, while the prefixes of nodes originally at positions $i + 1$, $i + 2$, and so on, lose exactly one node, namely $v$.

Instead of running grow-shrink from the empty set at every step as in BOSS, FLOP initializes grow-shrink with the previous parent set, that is, it continues from the result of grow-shrink for the previous prefix, now with one additional or one fewer node. The idea behind this strategy is that the parent set typically changes little when the prefix changes by just one node, and so this warm start makes parent selection far cheaper. Our implementation is given in Algorithm 3. Moreover, our grow-shrink does not insist on inserting or removing the single best parent with largest score improvement, making it *non-greedy*; it adds or removes any parent that improves the score, even if not maximally so. This eliminates the need for complicated grow-shrink tree caching as used in BOSS.

We show that the modified grow-shrink with warm start learns the restricted Markov boundary of a node $v$ with respect to a set $Z$ (Lam et al., 2022). This yields theoretical guarantees that the DAG learned by FLOP is the sparsest Markovian one for the considered order (Raskutti & Uhler, 2018).

**Definition 3.1.** *Let $P$ be a distribution over $X_1, \ldots, X_p$. The restricted Markov boundary of $X_v$ relative to a set $Z \subseteq \{X_1, \ldots, X_p\} \setminus \{X_v\}$, denoted by $M(v, Z)$, is defined as a set of nodes $M \subseteq Z$ such that a) $X_v \perp (Z \setminus M) \mid M$ and b) there exists no $M' \subset M$ such that $X_v \perp (Z \setminus M') \mid M'$.*

Under mild assumptions, the Markov boundary is unique (Verma & Pearl, 1988). As in GRaSP (Lam et al., 2022) and BOSS (Andrews et al., 2023), we learn it using BIC score improvements in place of conditional independence tests (Margaritis, 2003). In the sample limit, the local BIC score $\ell(X_v, X_{\mathrm{Pa}(v)} \cup \{X_u\})$ is smaller than $\ell(X_v, X_{\mathrm{Pa}(v)})$ if, and only if, $X_v \not\perp X_u \mid X_{\mathrm{Pa}(v)}$ (Koller & Friedman, 2009). We show that this asymptotic guarantee also carries over to the modified grow-shrink algorithm that starts from an arbitrary initial parent set instead of the empty set.

**Lemma 3.2.** *Let data set $D$ consist of $n$ i.i.d. observations of a probability distribution represented by a Bayesian network over variables $X_1, \ldots, X_p$. Then, in the large sample limit of $n$, grow-shrink finds the restricted Markov boundary of node $v$ relative to a set $Z \subseteq \{X_1, \ldots, X_p\} \setminus \{X_v\}$ when started with any initial set $P \subseteq Z$.*

*Proof.* This follows directly from the proof of correctness of the grow-shrink algorithm in (Margaritis, 2003) and its generalization to restricted Markov boundaries in (Lam et al., 2022). Assume that at the end of the grow-phase, the current set of parents is $P_{\mathrm{grow}}$. Thus, it holds that $X_v \perp X_u \mid P_{\mathrm{grow}}$ for all $X_u \in Z \setminus P_{\mathrm{grow}}$, or rephrased $X_v \perp Z \setminus P_{\mathrm{grow}} \mid P_{\mathrm{grow}}$. However, this would violate the

---

**Algorithm 3:** Non-greedy grow-shrink with the option to start from a previous parent set $P_{\text{prev}}$.

1  **function** `growShrink`($u$, $Z$, $D$, $P_{prev}$, $\ell_{prev}$, $\delta$)
2    **if** $\delta > 0$ **then** // $\delta$ is the node added to the candidate parents $Z$
3       $P_{\text{new}} := P_{\text{prev}} \cup \{\delta\}$
4       $\ell_{\text{new}} := $ `localScore`($u$, $P_{new}$, $D$) // score with $\delta$ added to parents
5       **if** $\ell_{new} < \ell_{prev}$ **then return** $\ell_{\text{prev}}$, $P_{\text{prev}}$ // return if no improvement
6    **else if** $\delta < 0$ **then**// $|\delta|$ is the node removed from the candidates $Z$
7       **if** $|\delta| \notin P_{prev}$ **then return** $\ell_{prev}$, $P_{prev}$ // return if $|\delta|$ was no parent
8       $P_{\text{new}} := P_{\text{prev}} \setminus \{|\delta|\}$
9       $\ell_{\text{new}} := $ `localScore`($u$, $P_{new}$, $D$)
10    **else** // If no $\delta$ is provided, run from scratch.
11       $(P_{\text{new}}, \ell_{\text{new}}) := (\emptyset,$ `localScore`($u$, $\emptyset$, $D$) $)$
12    **end**
13    `grow`($u$, $P_{new}$, $\ell_{new}$, $Z$, $D$)
14    `shrink`($u$, $P_{new}$, $\ell_{new}$, $Z$, $D$)
15 **end**

16 **function** `grow`($u$, $P$, $\ell$, $Z$, $D$)
17    **repeat**
18       **foreach** $v \in Z \setminus P$ **do**
19          **if** $\ell_{new} := $ `localScore`($u$, $P \cup \{v\}$, $D$) $< \ell$ **then** $(\ell, P) := (\ell_{\text{new}}, P \cup \{v\})$
20       **end**
21    **until** $P$ *is unchanged*
22 **end**

23 **function** `shrink`($u$, $P$, $\ell$, $Z$, $D$) ... // Analogous to grow, thus omitted.

---

uniqueness of the Markov boundary $M(v, Z)$ if it is not a subset of $P_{\text{grow}}$. This argument does not depend on the initial set $P$. The correctness of the shrink-phase is unchanged, too. $\qquad\square$

In addition to the warm start, we implement another optimization. We pass the node $v$ that we are either inserting to (coded as $+v$) or removing from (coded as $-v$) the prefixes into grow-shrink. If $v$ has been removed and was not part of $P_{\text{prev}}$, we immediately return $P_{\text{prev}}$. If node $v$ has been inserted to the prefix and does not increase the score when added to $P_{\text{prev}}$, we again immediately return $P_{\text{prev}}$. We show that these modifications preserve the guarantees above. In the sample limit, FLOP returns a Markovian DAG, that is, one that induces no additional conditional independencies.

**Theorem 3.3.** *Let data set $D$ consist of $n$ i.i.d. observations of a probability distribution represented by a Bayesian network over $X_1, \ldots, X_p$. In the sample limit of $n$, the CPDAG returned by FLOP is Markovian to $P$.*

*Proof.* This statement holds assuming that the grow-shrink procedure in FLOP finds a Markovian graph for each scored order. As the grow-shrink routines depend on the previous runs, we prove this by induction. Initially, a standard grow-shrink is run for the starting order (line 2 of Algorithm 1), which yields parent sets corresponding to its sparsest Markovian DAG (Raskutti & Uhler, 2018; Lam et al., 2022). Assume that the parent sets for the previous order have this property. By Lemma 3.2, the modified grow-shrink, if run fully, finds the restricted Markov boundary with respect to the prefix and thus yields parent sets of the sparsest Markovian DAG. It remains to show that the two early breaks in lines 5 and 7 of Algorithm 3 are correct, where the grow-shrink is not run.

If removed node $\delta$ was not part of the previous Markov boundary, clearly the Markov boundary remains unchanged for the reduced prefix, as neither the grow nor the shrink phase would add or remove a node. If added node $\delta$ does not increase the score for the enlarged prefix, this is the case, too. Thus, by Lemma 3.2, the DAG learned by FLOP is Markovian and the statement follows by the fact that the CPDAG of such a DAG is returned. $\qquad\square$

We remark that further modifications to FLOP in the subsequent sections do not change this result. As with BOSS, one can make FLOP asymptotically consistent, provably yielding the true graph in

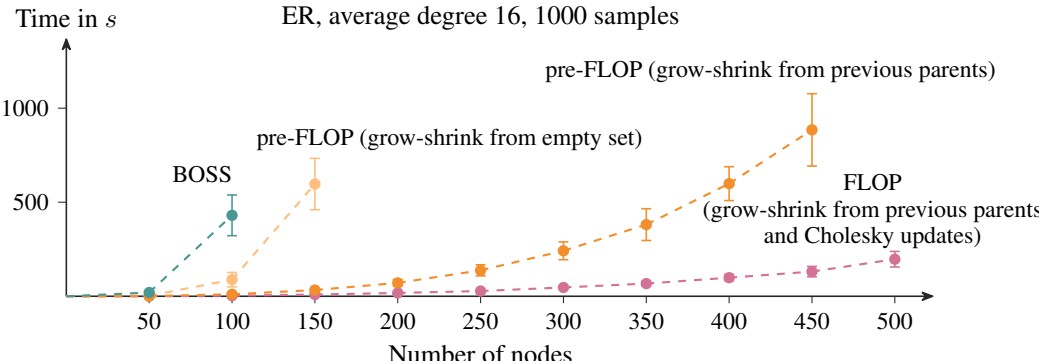

Figure 2: Run-time in seconds, averaged over 50 repetitions with standard-deviation error bars, for ER graphs with average degree 16, 1000 samples, and $\{50, 100, 150, \ldots, 500\}$ nodes.

the sample limit under the faithfulness assumption, by running the backwards phase of GES (Chickering, 2002) after termination of the local search. However, we refrain from this, since FLOP already reaches state-of-the-art finite-sample performance without it. More generally, any score-based discovery algorithm can be made asymptotically consistent by running another consistent algorithm, for example GES or PC, in parallel and returning whichever graph attains the better score.

## 3.2 DYNAMIC CHOLESKY UPDATES

We exploit the local structure of grow-shrink to avoid recomputing the local score from scratch at every step. Instead, we update the score from the previous parent set which at each step changes only by a single insertion or removal. This idea is generic and applies to any local search that adds or removes one edge at a time in a local search, not only to order-based methods.

In the multivariate normal setting, the local BIC score at node $u$ is $\ell(X_u, X_{\mathrm{Pa}(u)}) = n \log(\widehat{\mathrm{Var}}_D(X_u \mid X_{\mathrm{Pa}(u)})) + \lambda \ln(n)|\mathrm{Pa}(u)|$. Since the penalty depends only on the size of the parent set, the work in computing or updating the local score is in the likelihood term, in the estimated conditional variance of $X_u$. A direct way to compute this would be to invert a submatrix of the sample covariance matrix, but it is numerically more stable and faster to avoid the matrix inversion in favor of using Cholesky factorizations. As shown in Appendix B, the bottom right entry of the Cholesky factor of the covariance submatrix corresponding to $X_{\mathrm{Pa}(u)}$ and $X_u$ yields the square root of the conditional variance of $X_u$ given $X_{\mathrm{Pa}(u)}$.

Computing the Cholesky decomposition of a $k \times k$ matrix requires $(1/3)k^3$ floating-point operations. However, as discussed above, the submatrix which we Cholesky-factorize changes by adding or removing only a single row and column (corresponding to the added or removed parent node). We therefore update the Cholesky factor instead of recomputing it, using standard rank-one update and downdate routines (Gill et al., 1974; Golub & Van Loan, 2013). These updates require $O(k^2)$ floating-point operations, shaving off a factor $k$ compared to a fresh Cholesky decomposition. This run-time improvement proves advantageous for larger and denser graphs. These Cholesky updates are applicable to other score-based causal discovery algorithms, for example GES or other hill-climbers. To our knowledge, this speedup has not been described in prior causal discovery work.

## 3.3 RUN-TIME COMPARISON

In Figure 2, we compare the run-time of FLOP, which includes the two run-time improvements described in this section, with two ablated versions, termed *pre-FLOP* in the plot, the first one using neither optimization (thus only differing from BOSS by using *non-greedy* grow-shrink) and the second one only using the grow-shrink started at the previous parent set. For reference, we provide the run-time of the BOSS implementation in the `Tetrad` software package. The comparison with Tetrad is not apples-to-apples, since Tetrad is written in Java and multithreaded, while our code is Rust and single-threaded. The benchmark uses Erdős-Renyi graphs with average degree 16, ori-

ented according to a uniformly-random linear order (details on the simulation setup are provided in Section 5). Each run has a 30 minute time limit.

Both the modified grow-shrink and the Cholesky updates yield substantial run-time reductions. With both optimizations, FLOP is more than a factor 100 faster than BOSS for graphs with 100 nodes and scales to 500 nodes, whereas BOSS reaches the time limit for instances with 150 nodes. We note that accuracy of the discovered graphs is similar on these instances for both methods, both giving good, but not perfect results. In the following, we use the optimized FLOP and build on these speedups to further improve the quality of the found graphs.

## 4    IMPROVING THE ACCURACY OF ORDER-BASED SEARCH

This section presents two techniques to improve search accuracy. First, we replace random initial orders with a principled initial order construction putting strongly-correlated nodes next to each other, which is critical on directed paths in finite samples. Second, we use Iterated Local Search (ILS), which perturbs a found solution and restarts the local search, trying to escape local optima through additional compute. With these techniques, FLOP attains state-of-the-art accuracy in simulations.

### 4.1    INITIAL ORDER

Path graphs $x_1 \to x_2 \to x_3 \to \cdots \to x_p$ are challenging instances for order-based methods, which, to our knowledge, have not been previously discussed in this context before. On the left of Figure 3, we compare different algorithms on path graphs with 50 nodes. FLOP with a random initial order (FLOP$_0^{\mathrm{rand}}$), and BOSS are in fact the worst-performing of all methods. A reason for this are far-apart ancestor-descendant pairs with very weak marginal dependence, for which the grow-shrink procedure may fail to add edges, resulting in non-Markovian DAGs in finite samples. For example, if $x_i$ and $x_j$ with $i \ll j$ appear first in the order, grow-shrink should, irrespective of the remaining order, make $x_i$ a parent of $x_j$ for it to yield a Markovian DAG since $x_i$ and $x_j$ are marginally dependent. However, the dependence between $x_i$ and $x_j$ may be too small for grow-shrink to pick up on in finite samples.

As a remedy, we build the initial order so that strongly correlated nodes are adjacent, facilitating grow-shrink to find a Markovian graph. To build the order, we start with the two most correlated nodes and append, at each step, the variable that can be best explained by variables already placed in the order, that is, the one with the smallest residual variance when regressed onto the nodes in the order. We standardize the data beforehand to avoid scale artefacts. We compute this order efficiently, by iteratively constructing a Cholesky decomposition of the covariance matrix choosing the next node in the order according to their residual variance (see Appendix B). On 50-node paths (Figure 3, left), FLOP with this initial order has an average SHD on-par with PC and GES, the best performing algorithms on these instances.

### 4.2    ITERATED LOCAL SEARCH

Iterated Local Search (ILS) is a classic metaheuristic in discrete optimization (Lourenço et al., 2018) that has been used in previous score-based search over DAGs and CPDAGs (Liu et al., 2023; Nazaret & Blei, 2024). It is a generic strategy that combines local search with perturbations to escape local optima: Run local search to a local optimum, perturb the best solution seen so far, then rerun local search starting from this perturbation; repeat. In principle, this procedure can be repeated indefinitely.

For FLOP, the first local search starts from the initial order constructed as described in the previous section. After that, the starting order for the next local search is obtained by perturbing the best-found order by $k$ random swaps of two (not necessarily adjacent) elements. The idea being, that orders near local optima are better starting points than fully random ones. We set $k = \ln p$ by default, which we found to yield robust results balancing moving far enough to escape while staying in a promising basin.

On dense Erdős-Renyi graphs with 25 nodes and an average degree of 16 (Figure 3, right), increasing the number of restarts of the local search (zero restarts amount to one local search, $x$ restarts to $x$ perturbations and new local searches after that), consistently improves FLOP's accuracy. With

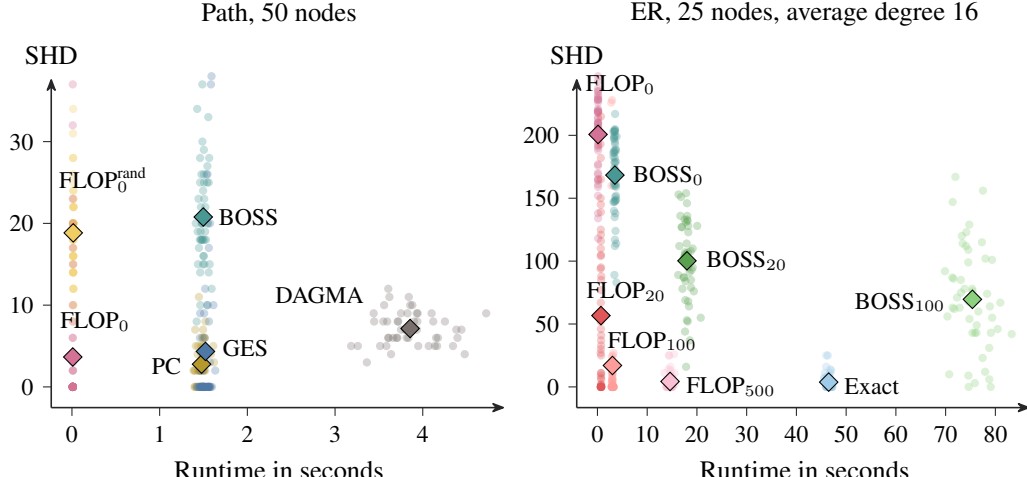

Figure 3: Run-time plotted against SHD on paths with 50 nodes for 1000 samples (left) and ER graphs with 25 nodes and average degree 16 for 50,000 samples (right). For the path graph, $FLOP_0$ finds the target graph in 72% of instances, PC in 32%, GES in 66% and the remaining algorithms in none; for the ER graphs, $FLOP_{20}$ does so in 26% of cases, $FLOP_{100}$ in 50%, $FLOP_{500}$ in 56%, Exact in 58%, $BOSS_{100}$ in 4% and the remaining algorithms in none.

500 restarts, FLOP matches the exact score-based algorithm while having a faster run-time (the exact score-based algorithm implements the method by Silander & Myllymäki (2006) and uses multithreading). We compare against BOSS with full random restarts, that is $x$ restarts mean $x + 1$ independent runs of BOSS and returning the best-found solution. This is computationally heavy and yields substantially smaller gains than FLOP's ILS restarts.

ILS is an integral part of the FLOP algorithm. When calling FLOP, the user needs to specify either the number of restarts of the local search or a time limit, and the solver runs ILS until the budget is exhausted. This emphasizes the trade-off between run-time and accuracy inherent to score-based causal discovery, but effectively ignored by the structure learning community with its focus on one-shot heuristic algorithms.

## 5 SIMULATIONS

We empirically compare FLOP to other causal discovery methods. For Figure 1, we generate Erdős-Renyi (ER) graphs with 50 nodes and average degree 8. We also consider scale-free (SF) graphs with density parameter $k = 4$, generated by starting with a star graph of $k + 1$ nodes and adding further nodes by preferential attachment to $k$ existing nodes, and DAGs from the bnlearn repository (Scutari, 2010), such as the Alarm network (Beinlich et al., 1989). We orient all graphs according to linear orders drawn uniformly at random. For each graph, we generate 1000 samples from a linear additive noise model with Gaussian noise (with mean 0 and variance uniformly drawn from $[0.5, 2.0]$) and edge coefficients drawn uniformly from $[-1, -0.25] \cup [0.25, 1]$. Each setting is repeated for 50 random instances.

In addition to FLOP and BOSS, we run PC (Spirtes et al., 2000) as a classical constraint-based method, GES (Chickering, 2002) as a traditional score-based algorithm, and DAGMA (Bello et al., 2022) as a gradient-based continuous optimization method. For BOSS, PC, and GES, we rely on the implementation in Tetrad (Ramsey et al., 2018) through `causal-cmd` version 1.12.0, for DAGMA we use version 1.1.0 of the authors' implementation. The algorithms are run on a machine with 256GB of RAM and an AMD Ryzen Threadripper 3970 CPU with 32 cores. We make no restrictions on the number of threads the implementations may use (FLOP only uses a single thread, whereas the other algorithms exploit multithreading) and report the wall-clock time of their execution. We use standard parameters in the literature, setting $\lambda_{BIC} = 2$ for the BIC-based algorithms (for a motivation of a higher penalty parameter than prescribed by the standard BIC, see Foygel & Drton, 2010),

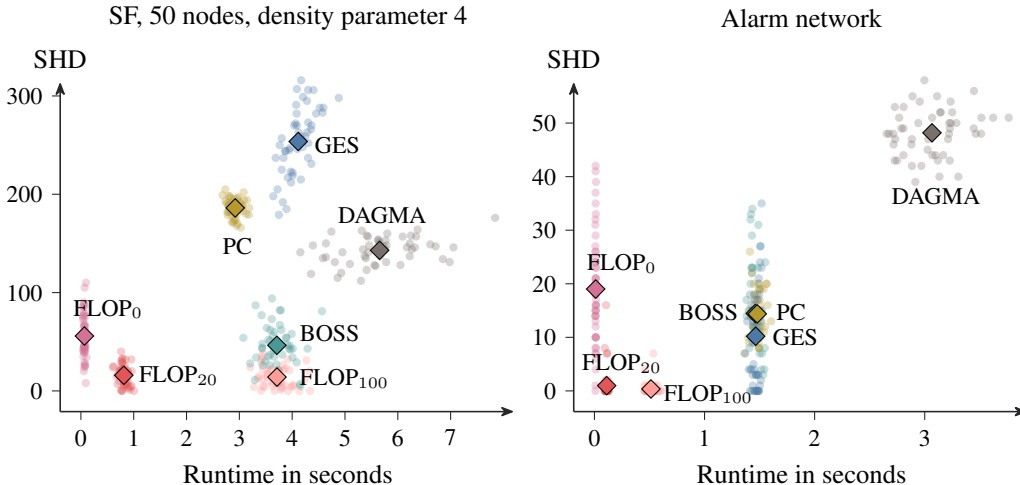

Figure 4: Run-time plotted against SHD on SF graphs (left) and the Alarm network, consisting of 37 nodes and 46 edges, (right), both for 1000 samples. For the SF graphs, $\text{FLOP}_{20}$ finds the target CPDAG in 6% of cases, $\text{FLOP}_{100}$ in 10%, the remaining algorithms in none; for the Alarm network, $\text{FLOP}_0$ does so in 2% of cases, $\text{FLOP}_{20}$ in 74%, $\text{FLOP}_{100}$ in 82%, BOSS in 6%, GES in 16%, DAGMA and PC in none.

$\alpha = 0.01$ for PC and $\lambda_{\text{DAGMA}} = 0.02$. As metric of accuracy, we report the Structural Hamming Distance (SHD) for CPDAGs, that is, the number of node pairs with differing edge relations in the compared graphs. If a method, such as DAGMA, returns a DAG, we first compute the corresponding CPDAG and compare this to the CPDAG of the true DAG (as we generally consider assumptions where only the CPDAG is identifiable). For some settings, we also report the Ancestor Adjustment Identification Distance (AID), measuring the mistakes when using the learned instead of the true CPDAG for the downstream task of causal effect identification (Henckel et al., 2024). For the PC algorithm, which does not always return a graph satisfying the invariants of CPDAGs, such as acyclicity, we report the AID only on runs that produced a valid CPDAG.

Figure 1 shows run-time versus SHD (lower left is better). On SF graphs, the order-based algorithms clearly outperform PC, GES, and DAGMA; FLOP with ILS improves further. Even with 100 ILS restarts, FLOP's run time is comparable to BOSS. On the Alarm network instances, the improvements through ILS are even more apparent, and with it FLOP obtains near-perfect results.

Graphs returned by FLOP achieve a lower SHD than competing score-based methods due to better optimization of the BIC score. This is shown in Figure 5, where we report the BIC score difference to the ground-truth DAG. Generally, the results look qualitatively similar to the SHD plots for the presented settings. However, for the SF graphs, it can be seen that the BIC, e.g., for $\text{FLOP}_{20}$ is close to zero, whereas the SHD for many instances lies clearly above zero. In fact, for a majority of runs, the BIC score of the graph found by $\text{FLOP}_{20}$ is even (slightly) better than the BIC score of the ground-truth graph showing that the global BIC optimum does not identify the ground truth in these cases.

We also evaluated the DAGMA loss function with MLE parameters fitted to the graph returned by FLOP and observed this to produce a lower loss compared to the graph and parameters returned by DAGMA itself. This casts doubt on the idea that gradient-based methods relying on differentiable DAG-constraints have an inherent advantage in optimizing their target score compared to discrete search. While these methods may offer other benefits, our results suggest that those likely come from aspects other than optimization quality.

## 6 DISCUSSION

In score-based causal discovery, two questions arise: (1) Is the true graph score-optimal? and (2) Can we find a score-optimal graph? Here we introduce FLOP, an efficient discrete optimization

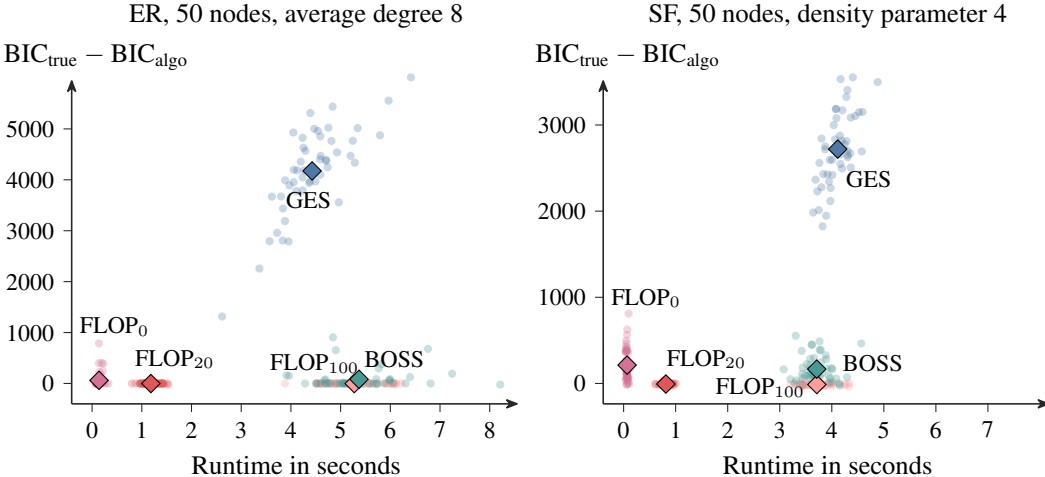

Figure 5: Run-time plotted against the BIC difference to the ground-truth graph for ER graphs on the left and SF graphs on the right. For the ER graphs, $FLOP_0$ finds a graph with better or equal BIC score than the true graph in 48% of cases, $FLOP_{20}$ and $FLOP_{100}$ in 84% of cases, BOSS in 52% of cases and GES in 0% cases. For the SF graphs, $FLOP_0$ finds such a graph in 6% of cases, $FLOP_{20}$ in 76% of cases, $FLOP_{100}$ in 94% of cases, BOSS in 6% of cases and GES again in 0% of cases.

algorithm to search for graphs minimizing the Gaussian BIC score. Indeed, when the true graph of a linear ANM has the globally optimal Gaussian BIC, FLOP typically recovers this graph or outputs one very close to it. This is further supported by simulations in Appendix C including uniform noise, unstandardized data, data based on an adaptation of the Onion method (Andrews & Kummerfeld, 2024), and real-world networks from bnlearn (Scutari, 2010). Across these settings, FLOP attains state-of-the-art accuracy, typically achieving better BIC and lower SHD in a fraction of the run-time of competing methods. When assumptions are violated (Appendix C.7-C.9) or sample sizes are too small for asymptotic guarantees to hold (Appendix C.4 and Figure 11 in Appendix C.6), FLOP still optimizes the Gaussian BIC as intended and finds graphs with better BIC score than the ground truth, but graph recovery suffers because the scoring criterion does not identify the true graph.

These results highlight that it is reasonable to revisit and embrace discrete search for causal structure learning. FLOP often finds graphs that are score-optimal or score better than the target graph and makes the link between accuracy and speed explicit: A computationally efficient search permits more exploration through iterated local search and that leads to better-scoring graphs. Our findings also recalibrate what is considered hard. First, ER graphs with 50 nodes and about 200 edges are often presented as challenging, yet for linear ANMs order-based discrete search solves them reliably and quickly. Second, on widely used linear benchmarks, the discrete optimization of the Gaussian BIC is feasible, not a bottleneck, and often more efficient and reliable than other optimization methods. Instead, key challenges in causal discovery lie in designing and selecting appropriate scoring criteria that identify the true graph as score-optimal not only asymptotically but with high probability also on finite samples.

At the same time, advancing causal discovery in practice remains difficult even on small graphs, since the ground truth is rarely known and assumptions are violated. It has been feasible for decades to find a global BIC optimum with exact exponential-time search up to roughly 30 variables (Koivisto & Sood, 2004; Silander & Myllymäki, 2006). FLOP extends strong BIC optimization to substantially larger graphs, but that does not make the practical problems go away. Our work shifts the attention away from inflated combinatorial hardness rhetoric and from a misattributed gap between asymptotic theory and observed finite-sample performance, toward the immense challenges causal discovery faces outside of synthetic benchmarks (Reisach et al., 2021; Göbler et al., 2024; Mogensen et al., 2024; Brouillard et al., 2025; Gamella et al., 2025; Gururaghavendran & Murray, 2025; Jørgensen et al., 2025).

ACKNOWLEDGMENTS

Marcel Wienöbst thanks Kenneth Langedal for fruitful discussions and introducing him to iterated local search. Sebastian Weichwald was supported by a research grant (0069071) from Novo Nordisk Fonden. This research was supported by the Pioneer Centre for AI, DNRF grant number P1.

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

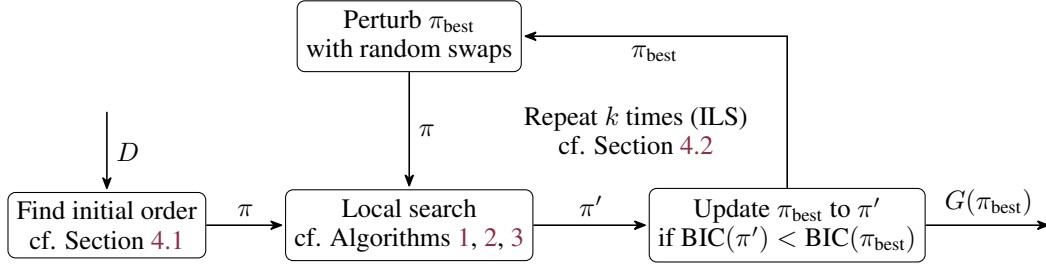

Figure 6: Visualization of the general control flow in the FLOP algorithm.

## A  HIGH-LEVEL DESCRIPTION OF THE FLOP ALGORITHM

As an overview of the FLOP algorithm, we provide Figure 6, describing the high-level control flow
of FLOP. The algorithm begins by computing an initial order for the given data set $D$. Afterwards,
the ILS loop starts with a local search aiming to improve the order $\pi$ through reinsertions. Reinsertions
are done by moving a node $v$ to its locally optimal position until no further improvements are
possible (Algorithm 1). This optimal reinsertion is computed as described in Algorithm 2, which
relies on the grow-shrink described in Algorithm 3 for updating the parents. This grow-shrink starts
at the previous parent set and uses efficient Cholesky updates for scoring as described in Section 3.2.

After the local search completes, the best found graph/order is updated ($\pi_{\text{best}}$ in Figure 6) if the score
is lower than the previous best. This best-scoring order found thus far is then perturbed as described
in Section 4.2 and the perturbed copy is then used as the starting point for the next local search. We
note that this procedure ensures that more ILS iterations can only improve the score of $\pi_{\text{best}}$ and thus
of the returned graph $G(\pi_{\text{best}})$ because $\pi_{\text{best}}$ is only updated if the local search after the perturbation
yields a better scoring order.

We note that the reinsertion-based local search follows the general principle of the BOSS algorithm,
which consists of reinserting nodes at their locally optimal position until no further (local) improvements
are possible. However, FLOP and BOSS differ in the following key aspects:

- The grow-shrink procedure of BOSS starts with the empty set. Moreover, it is a greedy
  grow-shrink that always inserts or removes the node with the largest local score improvement.
  In contrast, FLOP accepts any improving insertion in the grow-, and any improving
  removal in the shrink-phase. For the implementation of grow-shrink BOSS relies at its core
  on an intricate data structure called grow-shrink trees, which FLOP avoids. Overall, this
  allows FLOP to obtain a better run-time performance compared to BOSS.

- FLOP uses Cholesky updates for efficient iterative scoring during the local search and, in
  particular, in the grow-shrink routine. This yields further run-time gains.

- FLOP makes use of an iterated local search (ILS) that allows spending more compute for
  improved BIC optimization. As more ILS restarts can never yield worse scoring graphs,
  this effectively trades off compute with accuracy. Due to FLOPs run-time improvements
  this yields a free lunch with regard to accuracy gains.

- BOSS starts the local search with a random order. This leads to performance deteriorations
  on path instances as shown in Section 4.1. In contrast, FLOP explicitly constructs the initial
  order to avoid such problems.

## B  CHOLESKY DECOMPOSITION OF THE COVARIANCE MATRIX

Let $X_1, \ldots, X_p$ be real-valued centered random variables with finite second moments and with
full-rank covariance matrix $\Sigma = [\text{Cov}(X_r, X_c)]_{r,c \in \{1,..,p\}} \in \mathbb{R}^{p \times p}$, that is, for all $j \in \{1, ..., p\}$,
$\mathbb{E}(X_j) = 0$ and $0 < \Sigma_{j,j} = \text{Var}(X_j) < \infty$; further, $\Sigma$ is symmetric positive definite and admits a
unique Cholesky factorization $\Sigma = LL^\top$ with $L$ lower triangular and strictly positive diagonal.

For $j \in \{1, ..., p\}$, let $\widehat{X}_j$ denote the best linear predictor of $X_j$ from its predecessors $X_1, ..., X_{j-1}$, that is, the ordinary least squares projection.

Then for all $j \in \{1, ..., p\}$,

$$L_{jj}^2 = \mathrm{Var}(X_j - \widehat{X}_j),$$

that is, $L_{jj}$ is the standard deviation of the least squares residuals when linearly regressing $X_j$ onto its predecessors.

To obtain the statement, fix $j \in \{1, \ldots, p\}$. Block-partition the leading $j \times j$ principal submatrices of $\Sigma$ and $L$:

$$\Sigma_{1:j,\,1:j} = \begin{pmatrix} \overbrace{\underbrace{\Sigma_{1:(j-1),\,1:(j-1)}}_{s^\top}}^{\Sigma'} & \overbrace{\underbrace{\Sigma_{1:(j-1),\,j}}_{c}}^{s} \\ \Sigma_{j,\,1:(j-1)} & \Sigma_{j,j} \end{pmatrix}, \qquad L_{1:j,\,1:j} = \begin{pmatrix} \overbrace{\underbrace{L_{1:(j-1),\,1:(j-1)}}_{r}}^{L'} & 0 \\ L_{j,\,1:(j-1)} & \underbrace{L_{j,j}}_{\ell} \end{pmatrix}.$$

From $\Sigma = LL^\top$ we get the block identities

$$\Sigma' = L'L'^\top, \qquad s = L'r^\top, \qquad c = rr^\top + \ell^2.$$

Since $L'$ is full rank, the second identity gives $r^\top = L'^{-1}s$, hence

$$rr^\top = s^\top \left(L'^{-\top}L'^{-1}\right)s = s^\top \left(L'L'^\top\right)^{-1}s = s^\top \Sigma'^{-1}s.$$

Substituting into $c = rr^\top + \ell^2$ yields

$$\ell^2 = c - s^\top \Sigma'^{-1}s.$$

On the other hand, the centered OLS problem

$$\min_{a \in \mathbb{R}^{j-1}} \mathbb{E}\left[(X_j - a^\top X_{1:(j-1)})^2\right]$$

has normal equations $\Sigma'a^\star = s$, so $a^\star = \Sigma'^{-1}s$ is the unique minimizer, $\widehat{X}_j = s^\top \Sigma'^{-1}X_{1:(j-1)}$, and the minimal mean squared error is

$$\begin{aligned} \mathrm{Var}(X_j - \widehat{X}_j) &= \mathrm{Var}(X_j) + \mathrm{Var}(\widehat{X}_j) - 2\,\mathrm{Cov}(X_j, \widehat{X}_j) \\ &= c + s^\top \Sigma'^{-1}\,\mathrm{Var}(X_{1:(j-1)})\,\Sigma'^{-1}s - 2s^\top \Sigma'^{-1}\,\mathrm{Cov}(X_j, X_{1:(j-1)}) \\ &= c + s^\top \Sigma'^{-1}\Sigma'\Sigma'^{-1}s - 2s^\top \Sigma'^{-1}s \\ &= c - s^\top \Sigma'^{-1}s \end{aligned}$$

This equals $\ell^2$, that is, $L_{jj}^2 = \mathrm{Var}(X_j - \widehat{X}_j)$, as claimed.

We remark that if $X_1, ..., X_p$ are jointly Gaussian, then $\mathrm{Var}(X_j - \widehat{X}_j) = \mathrm{Var}(X_j \mid X_{1:(j-1)})$.

## C  FURTHER BENCHMARK SETTINGS

In this section, we consider further benchmark settings to investigate the stability of FLOP under different graph and data generation procedures. If not specified otherwise, we consider ER graphs with 50 nodes and average degree 8, and 1000 samples being drawn from the underlying linear additive noise model. We also consider settings where the assumptions of FLOP with a linear Gaussian BIC are (potentially) violated, such as uniform noise (Subsection C.1) and non-linear relations (Subsection C.7) as well as semi-synthetic (Subsection C.8) and real-world data (Subsection C.9). In some settings, such as the uniform-noise case this leads to no performance degradation, whereas in others it leads to significantly higher SHDs compared to settings where the assumptions are satisfied. We observe that in these settings, FLOP still reliably optimizes the BIC, meaning the performance of FLOP in large parts depends on how well the scoring criterion is suited to the data. Hence, practitioners need to be careful that the assumptions hold when applying FLOP. Moreover, this shifts the focus in research from designing optimization algorithms towards the development of efficient and practical scoring criteria (see also the discussion in Section 6).

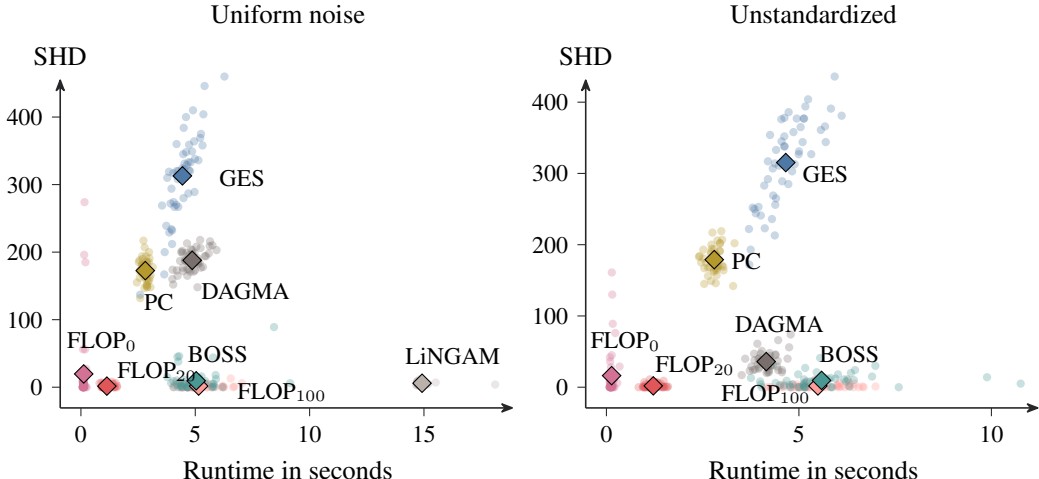

Figure 7: Run-time against SHD for data sampled with uniform instead of Gaussian noise on the left and for unstandardized data on the right (both settings are based on ER graphs with 50 nodes and average degree 8 with 1000 samples drawn). In the uniform noise case, $FLOP_0$ and BOSS find the target CPDAG in 34% of cases, $FLOP_{20}$ and $FLOP_{100}$ in 54% of the cases, the remaining algorithms in none. On unstandardized data, BOSS finds the target CPDAG in 22% of cases, $FLOP_0$ in 34% and $FLOP_{20}$ and $FLOP_{100}$ in 54% of cases, the remaining algorithms in none.

## C.1 UNIFORM NOISE

To check the performance of FLOP (using the Gaussian BIC to learn the CPDAG underlying a linear ANM) under non-Gaussian noise, we generate data with noise sampled uniformly from $[-1, 1]$. As the plot on the left of Figure 7 shows, there is no performance degradation (of any algorithm). Moreover, we compared the methods to DirectLiNGAM (Shimizu et al., 2011), which is based on identifiability theory for non-Gaussian noise. DirectLinGAM gets low SHD on these instances, but in the 50 repetitions never recovered the ground truth.

## C.2 RAW DATA

To avoid varsortability of the instances (Reisach et al., 2021), we typically standardize the data in the benchmarks as mention in Section 5. As an exception, we consider instances with unstandardized data on the right of Figure 7. As expected, we find that DAGMA performs significantly better than in the standardized settings. The performance of the other algorithms does not vary significantly. We also note that in FLOP we choose to always standardize the data to obtain a scale-invariant algorithm.

## C.3 DAO DATA

We also consider the DAG-adaptation of the Onion method (Andrews & Kummerfeld, 2024) as a way to sample data from an ANM. This method has been proposed to avoid artefacts in the data, such as R2-sortability (Reisach et al., 2023), which could be inadvertently or explicitly exploited to game benchmarks. In line with the results by Andrews & Kummerfeld (2024), we find that this sampling method yields harder-to-identify instances, with FLOP nonetheless performing best; however, all methods produce SHDs greater than 50, as shown in the left panel of Figure 8. This may be caused by weak causal relationships or (near)-faithfulness violations in the data. It is, however, not a failure in the optimization, as we observed that, for FLOP and other score-based algorithms, the BIC score of the learned graph was better than the one of the ground truth, suggesting non-identifiability of the true CPDAG under the BIC for the provided number of samples.

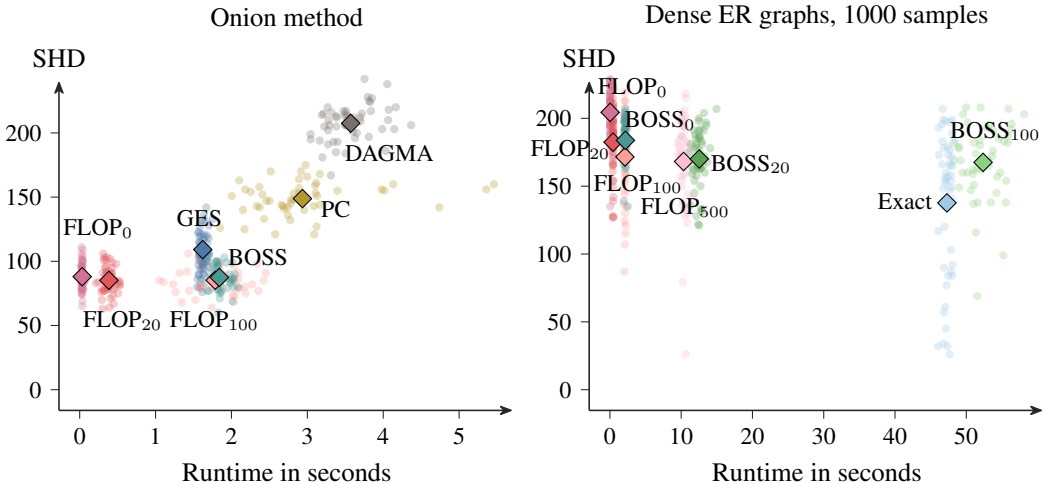

Figure 8: Run-time against SHD for data sampled with the DAG-adaptation of the Onion method on the left (again on ER graphs with 50 nodes and average degree 8 with 1000 samples drawn) and for dense ER graphs (25 nodes, average degree 16) with 1000 samples generated in the standard way on the right. In both settings, none of the algorithms ever recover the target CPDAG.

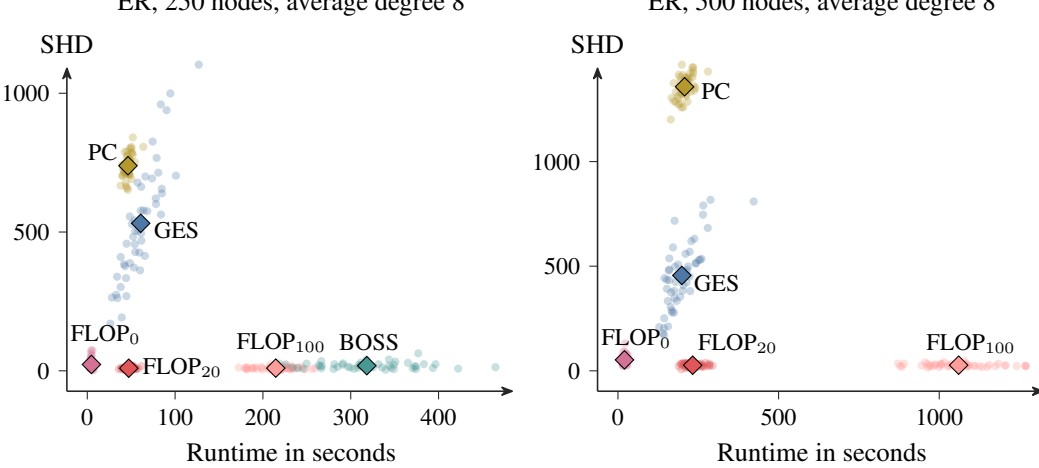

Figure 9: Run-time against SHD for ER graphs with 250 nodes on the left and with 500 nodes on the right (average degree 8 and 1000 samples drawn). BOSS times out on the latter instances. In both settings, none of the algorithms ever recover the target CPDAG.

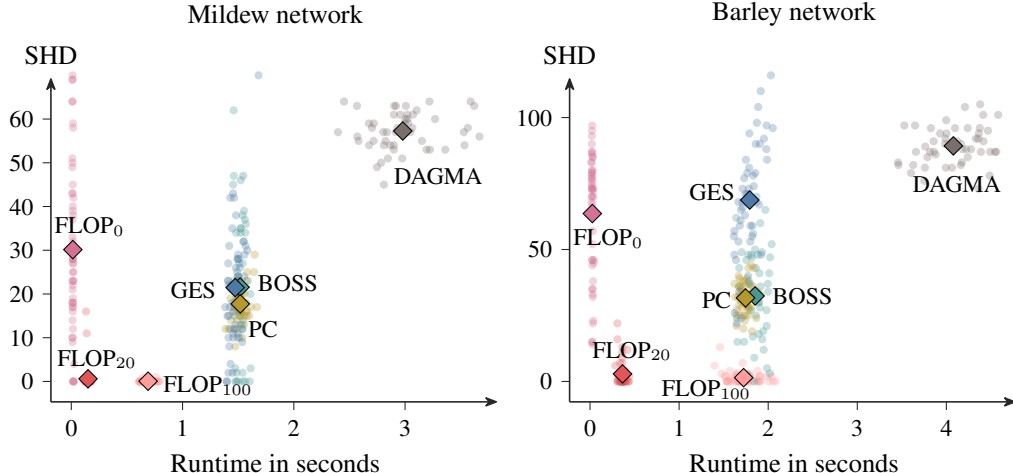

Figure 10: Run-time against SHD for the Mildew network on the left (Jensen & Jensen, 1996), which consists of 35 nodes and 46 edges, and the Barley network on the right (Scutari, 2010), which consists of 48 nodes and 84 edges. For the Mildew network, BOSS finds the target CPDAG in 2% of cases, $FLOP_{20}$ in 48% and $FLOP_{100}$ in 52% of the cases, the remaining algorithms in none. For the Barley network, GES finds the ground truth in 4% of cases, $FLOP_0$ in 8%, BOSS finds the target CPDAG in 12%, $FLOP_{20}$ in 90% and $FLOP_{100}$ in 94% of cases, PC and DAGMA in none.

### C.4 DENSE ER GRAPHS

In the main paper, we considered dense ER graphs (25 nodes and average degree 16) in a setting with 50,000 samples. Due to the denseness of the graph such a large amount of samples is necessary to identify the target graph. Here, we show the performance of the algorithms for significantly fewer samples, namely 1000 samples, as in the other simulations. As can be seen on the right of Figure 8, FLOP still performs quite well, however, the algorithms are much closer with regard to the SHD.

We note that we again compared the BIC score of the graph returned by FLOP with the ground-truth graph as well as the other algorithms. We found that FLOP and the exact algorithm consistently found graphs with a better BIC score than the ground truth or the other discovery algorithms' output graphs. This illustrates that the sample size is too small for the BIC score to reliably identify the true CPDAG.

Another thing to note is that compared to the setting with 50000 samples in the main text, both BOSS and FLOP run faster on instances with 1000 samples, whereas there is no noticeable difference for the exact algorithm. The reason for this increased run-time for larger samples sizes is that the BIC penalizes edges stronger for smaller sample sizes with the penalty term growing with $\ln n$ and the likelihood term proportional with $n$. Thus, intermediate graphs in the search are typically denser for high-sample settings, which increases the computational effort.

### C.5 LARGE ER GRAPHS

We also report the accuracy for large ER graphs with 250 and 500 nodes and average degree 8 in Figure 9. Here, DAGMA does not terminate within the time limit for either instances and BOSS does not for the graphs with 500 nodes. Overall, similar accuracy results as before can be observed though notably PC appears to get worse with an increased number of variables in comparison with GES.

### C.6 BNLEARN GRAPHS

In addition to the random graphs, we also consider real-world networks from the bnlearn repository, namely the Mildew (Jensen & Jensen, 1996), Barley and the Pathfinder network (Heckerman et al., 1992). All three are too large such that exact score-based algorithm based on dynamic programming

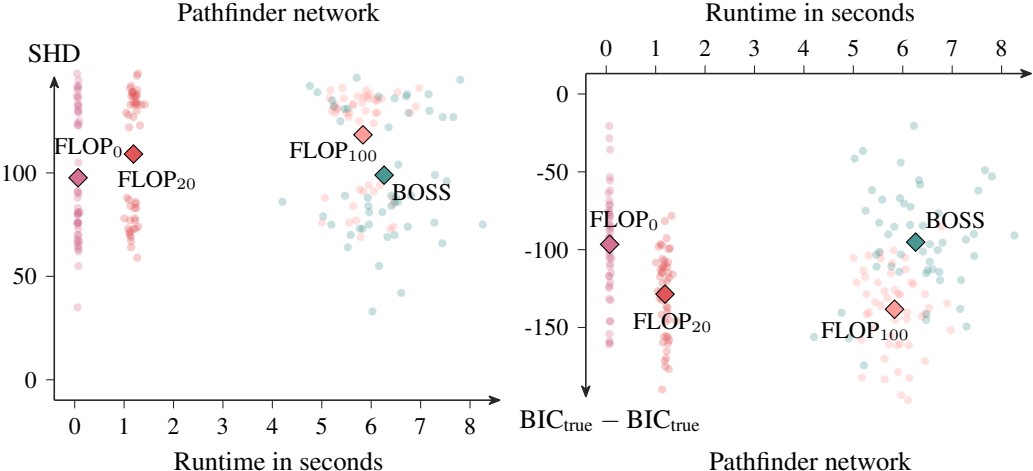

Figure 11: Run-time against SHD for the Pathfinder network, which consists of 109 nodes and 195 edges, on the left. On the right, the BIC score difference compared to the ground-truth graph for the Pathfinder network. None of the algorithms ever recover the target CPDAG.

could be used, with Mildew consisting of 35 nodes and 46 edges, Barley of 48 nodes and 84 edges, and Pathfinder of 109 nodes and 195 arcs. In all cases, we generate the data synthetically in the same manner as before. For Mildew and Barley on the left and right of Figure 10, FLOP performs significantly better than other methods and, in particular, that the ILS is needed to get close-to-perfect accuracy on these instances. For Pathfinder on the left of Figure 11, PC, GES and DAGMA do not terminate within the time limit of 30 minutes. Here, $FLOP_0$ and BOSS yield roughly similar SHD. However, with an increasing number of ILS iterations, the SHD gets worse for $FLOP_{20}$ and $FLOP_{100}$. To analyze this behaviour further, we show the BIC score difference to the ground-truth DAG on the right of Figure 11. Indeed, all reported methods yield better BIC scores than the true DAG and ILS does find even better-scoring graphs, which, in this case, are further from the ground truth. Again, faithfulness violations promoted by the underlying graph structure may be the issue here, even though closer investigations are needed.

## C.7 NON-LINEAR DATA

As settings where the linear Gaussian BIC is misspecified, we consider non-linear data generated from a randomly initialized multi-layer perceptron (MLP) with a single hidden layer of size 100 and sigmoid activation, as described in Appendix C.2.2 in (Bello et al., 2022) and from sampled Gaussian process regressions with a unit bandwidth RBF kernel as proposed in (Rolland et al., 2022). In both settings, the ground-truth DAG is generated by orienting an ER graph with 25 nodes and average degree 4, thus containing on average 50 edges, according to a linear order that is drawn uniformly at random. We consider the same algorithms as before with the same parameter choice and score. They are hence not tuned towards the non-linear setting. Additionally, we include the non-linear DAGMA algorithm from Bello et al. (2022). In Figure 12, we plot the SHD of each method contrasted with the BIC difference to the optimal BIC score (for $\lambda_{BIC} = 2$) for each of the algorithms (cases where PC does not return a valid CPDAG are omitted). As can be seen, the BIC optimum does not correspond to low SHD in both settings with GES, BOSS and $FLOP_{100}$ having similar performance, and the ground truth having suboptimal BIC scores. For the MLP setting, the non-linear version of DAGMA is the best method for graph recovery, however, in the GP setting it is not better than the other approaches. It is also the by far slowest method, taking over 5 minutes per instance. The assumptions of the LiNGAM algorithm are also violated by the non-linearities and it is clearly the worst-performing algorithm among the presented ones.

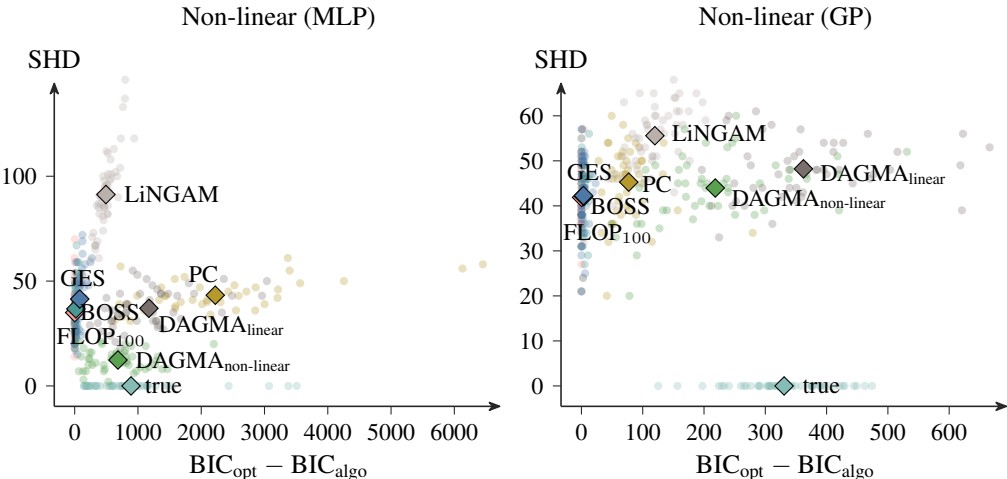

Figure 12: BIC score difference to the BIC optimum plotted against SHD for non-linear data generated based on MLPs with a single hidden layer on the left. On the right, non-linearities are generated from sampled Gaussian process regressions with a unit bandwidth RBF kernel. Both settings use ER graphs with 25 nodes and average degree 4, thus the ground truth contains 50 edges on average. In the MLP setting, $FLOP_{100}$ finds the BIC optimum in 44% of cases, BOSS finds it in 12% of cases and GES in 4% of cases. In the GP setting, $FLOP_{100}$ finds the BIC optimum in 72% of cases, BOSS in 16% of cases and GES in 30% of cases.

## C.8 CAUSALASSEMBLY DATA SET

We show the results on the causalAssembly dataset introduced by Göbler et al. (2024) in Figure 13. The ground-truth DAG consists of 98 nodes and 485 edges. We subsample 5000 observations with replacement from the data set 50 times and run the algorithms on this subsampled data. We exclude DAGMA and LiNGAM from the plots as they yield significantly larger SHD, which lies above 550, and take much longer than the competing algorithms, namely more than 30 seconds in the case of DAGMA and more than 200 seconds in the case of LiNGAM. The remaining algorithms return results of similar quality, with notable improvements through the ILS restarts that FLOP uses. These small improvements stem from better BIC score optimization as shown in the right plot. Here, the BIC difference to the true graph is reported and it is clear that all methods return graphs with much better BIC scores than that of the ground truth, suggesting score misspecification.

## C.9 SACHS DATA SET

We also evaluate the algorithms on the Sachs dataset (Sachs et al., 2005), which consists of 11 nodes, and compute the SHD with regard to the CPDAG of the ground truth consisting of 17 edges. We run each algorithm using the same hyperparameters as before on 50 bootstrap samples of the 853 observations in the data set. As result, we observed $FLOP_{20}$, BOSS and GES performing on par, all yielding an average SHD of 12.58. The other algorithms yield similar results, with DAGMA having the best performance with an average SHD of 11.7. The PC algorithm obtains an average SHD of 12.56 and LiNGAM an average SHD of 14.18.

## C.10 RESULTS FOR DIFFERENT PARAMETERS CHOICES

FLOP has two parameters that need to be chosen by the user. First, $\lambda_{BIC}$ scales the penalty term of the BIC and, second, the number of ILS restarts control the amount of compute that is invested. For the latter parameter, we have typically shown the simulation results for multiple choices, such as $FLOP_0$, when no restarts are performed, as well as $FLOP_{20}$ and $FLOP_{100}$ with 20 and a 100 restarts, respectively. We also note that more ILS iterations can only improve the BIC optimization.

For $\lambda_{BIC}$ on the other hand, we choose the value 2, which is the standard setting from the literature. As Foygel & Drton (2010) have shown, a larger value than 1 should be chosen to recover the structure

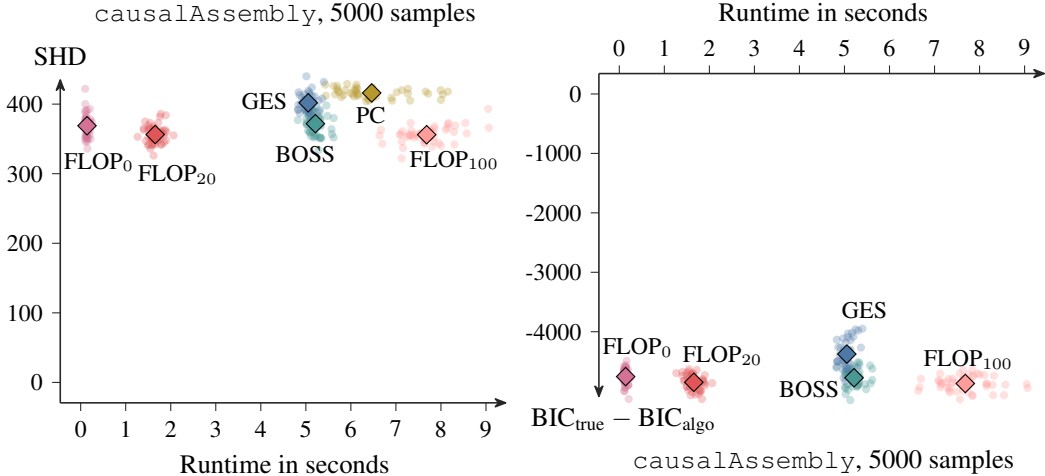

Figure 13: Run-time against SHD for the causalAssembly data on the left. On the right, the BIC score difference to the BIC of the ground truth is reported. LiNGAM and DAGMA are not included because both are considerably slower on these instances (with DAGMA needing more than 30 seconds per instance and LiNGAM more than 200 seconds) and obtain significantly worse SHD compared to the other methods, typically above 550 for both algorithms. As a point-of-reference, the ground-truth graphs consists of 485 edges, thus these two methods give worse SHDs than the empty graph. All methods optimizing the BIC, shown in the right graph, yield BIC scores clearly lower than that of the true graph, indicating score misspecification for the linear Gaussian BIC.

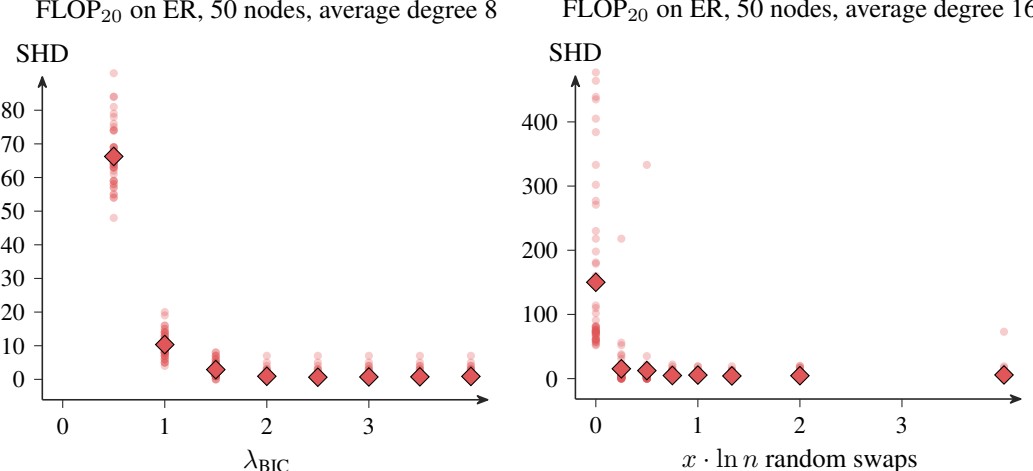

Figure 14: SHD for $\text{FLOP}_{20}$ with different choices of $\lambda_{\text{BIC}}$ on the left (ER graphs with 50 nodes and average degree 8) and with different factors $x$ controlling the number of swaps in a perturbation on the right (ER graphs with 50 nodes and average degree 16). For the $\lambda_{\text{BIC}}$, as explained by the derivation of the extended BIC (Foygel & Drton, 2010), values higher than 1 are needed on finite samples with 2 being a common choice. For the perturbations, it can be seen that many choices for the number of random swaps are effective (the exception being no perturbations, and thus no ILS at all, which is shown at $x = 0$, yielding an SHD in the hundreds for this setting), with outliers increasing for too few or too many swaps.

of graphical models, while any constant value guarantees asymptotic consistency. The results on the left of Figure 14 confirm this, showing that for $\lambda_{\text{BIC}}$ larger or equal to 2, graphs close to the ground truth are recovered by $\text{FLOP}_{20}$, while smaller choices of $\lambda_{\text{BIC}}$ yields spurious edges and thus a higher SHD.

Finally, for the ILS perturbations, FLOP defaults to $\ln p$ many random swaps. The software interface of FLOP does not allow tuning this parameter, as we found it to be a stable default choice. This is confirmed on the right of Figure 14, which runs FLOP with $x \cdot \ln p$ many random swaps for $x \in \{0, 1/4, 1/2, 3/4, 1, 4/3, 2, 4\}$. In the case that $x$ is set to zero, which corresponds to not running ILS, this yields an SHD that is often in the hundreds. Conversely, any of the positive choices of $x$ lead to good performance. The best results are obtained for $x$ between $3/4$ and $2$, while for the largest and smallest values of $x$ the number of outliers increases.

## C.11 ANCESTOR ADJUSTMENT DISTANCE

We show the Ancestor Adjustment Identification Distance (AID) as another metric for evaluating the learned graphs (Henckel et al., 2024). It effectively counts the number of mistakes one would make if one used the learned graph to select valid adjustment sets (using the ancestors of a node) instead of the ground-truth graph. Figure 15 shows the AIDs for a selection of the previous simulation results. We note that we omit data points where PC does not return a CPDAG (as is well-known to happen on finite samples). For example, on ER graphs with 500 nodes, the PC algorithm does not yield a single valid CPDAG.

## C.12 OTHER GRADIENT-BASED ALGORITHMS

Due to the choice of the least-squares loss function, for DAGMA and other popular gradient-based methods, one implicitly assumes perfect varsortability (or even equal noise variances in the underlying linear ANM) to recover the underlying DAG as the unique score-optimal DAG (Peters & Bühlmann, 2014; Reisach et al., 2021; 2023). Because we compare the methods on *standardized* data in our simulations (with the execption of Subsection C.2), this assumption is violated. In comparison, the GOLEM algorithm proposed by Ng et al. (2020) has a version that is developed specifically for the non-equal noise variance case. We ran this algorithm available in the `gCastle` package (version 1.0.4) with default parameters on the standard setting in this work (ER graphs with 50 nodes and average degree 8) and observed an average SHD of 176.6. This is slightly better than DAGMA with an SHD of 195.08 (which may also be partially due to different hyperparameter choices), but clearly worse than $\text{FLOP}_{100}$ with an average SHD of 1.34. The GOLEM algorithm also had a run-time of more than 200 seconds per instances, whereas running $\text{FLOP}_{100}$ took around 5 seconds per instances.

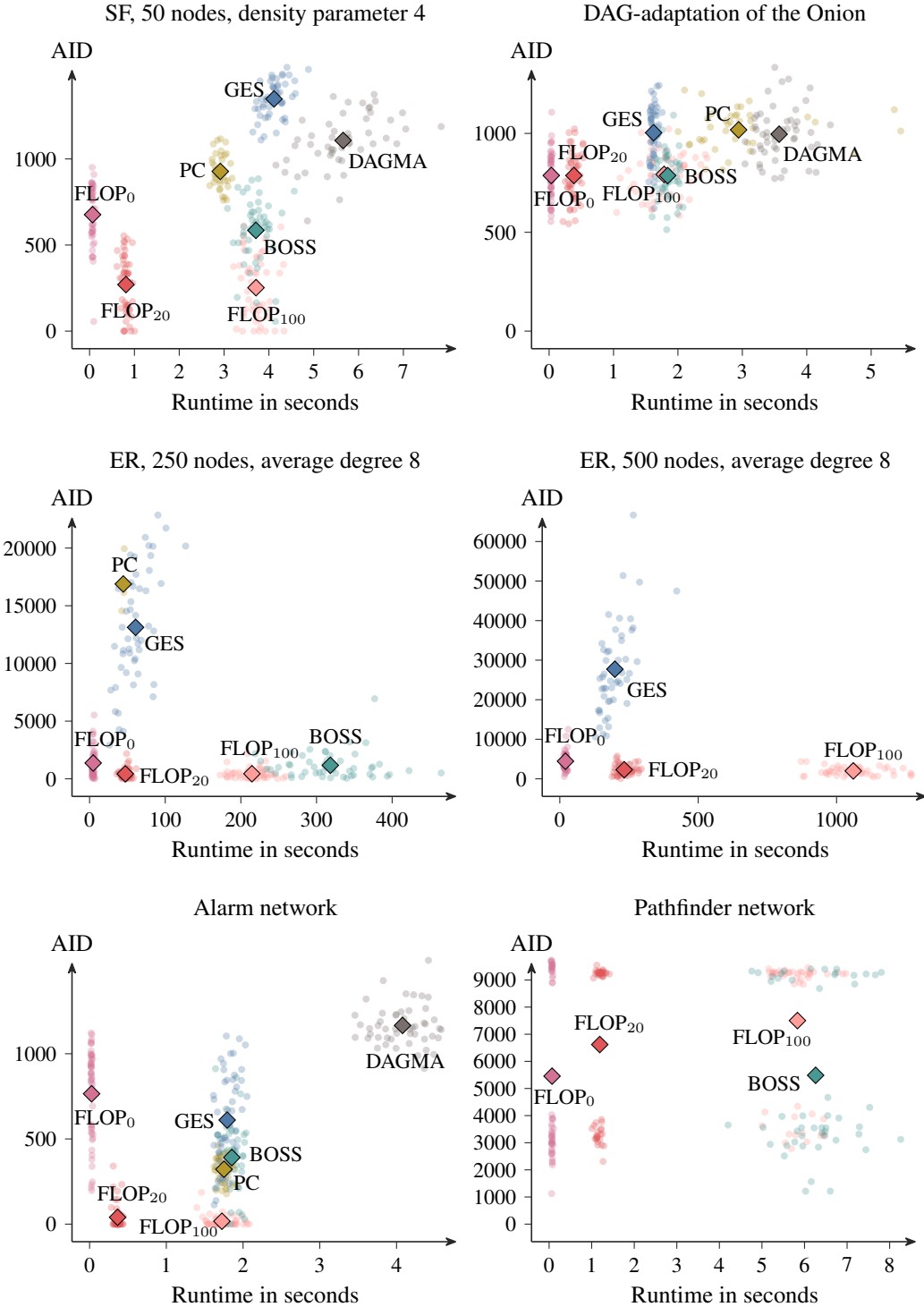

Figure 15: Run-time and AID for SF graphs (top left), data sampled by the DAG-adaptation of the Onion method (top right), ER graphs with 250 nodes (center left), ER graphs with 500 nodes (center right), the Alarm network (bottom left) and the Pathfinder network (bottom right).

