# OpenReview forum: "Embracing Discrete Search: A Reasonable Approach to Causal Structure Learning"
_ICLR.cc/2026/Conference — ICLR 2026 Poster_

### Official Review · Reviewer_UFtS · 2025-10-26

**Soundness:** 3
**Presentation:** 3
**Contribution:** 1
**Rating:** 2
**Confidence:** 3

**Summary:**

This paper introduces FLOP (Fast Learning of Order and Parents), a novel score-based, discrete search algorithm for causal structure learning for linear additive noise models (ANMs). FLOP builds upon an order-based search framework, similar to the BOSS algorithm, but introduces four key components that lead to significant gains in both speed and accuracy: warm-start parent selection, dynamic Cholesky updates, principled order initialization, iterated local search. Experiments across a wide range of benchmarks (ER, SF, Alarm, etc.) demonstrate that FLOP variants achieve state-of-the-art accuracy (measured by SHD and AID) while being orders of magnitude faster than competing methods like BOSS, GES, and DAGMA.

**Strengths:**

1. This paper is well-written and easy to follow.

2. By combining warm-start parent selection with dynamic Cholesky updates, FLOP achieves massive computational speed-up. Furthermore, this speed-up enables more exhaustive search, leading to higher accuracy.

3. The performance of FLOP is validated across many benchmarks.

**Weaknesses:**

1. The authors do not provide theoretical analysis on finite-sample performance of FLOP. I think this is a major weakness of this paper. First, the authors repeatedly emphasize the importance of finite-sample performance, e.g., "Finite-sample performance is a practical challenge, despite the mature identifiability theory and asymptotic guarantees", "one of the core issues of score-based methods in practice are finite-sample induced local optima". Second, the authors consider the simplest structural causal model where structural functions are linear, exogenous noises are Gaussian, and there is neither latent confounder nor selection bias.

2. The performance of FLOP is only validated on synthetic data. Although the authors consider real-world networks from the bnlearn repository, the data is still synthetic. This is not convincing for a paper whose primary contribution does not lie in theory.

3. The ILS perturbation strategy is defined as $k = \ln p$ random swaps of two elements. The paper simply states this is "robust" without an ablation study or sensitivity analysis.

**Questions:**

N/A

---

> ### Author Response · Authors · 2025-11-20
>
> We thank the reviewer for their feedback and appreciate the opportunity to clarify several points.
>
> __Detailed Responses__
>
> - __Finite-sample guarantees.__
> The focus of our paper is to devise and apply effective tools for improving the discrete search over orders and graphs to optimize scoring criteria such as BIC used in score-based causal discovery algorithms. While theoretical guarantees (of the score criterion) for finite-sample performance would be desirable, we deem this out-of-scope of our work.
>
> - __Synthetic data.__
> Despite the limitations of synthetic data benchmarking, we believe that it is an important and valuable contribution to report the performance improvements obtained across the community's standard causal discovery benchmarks when using FLOP to optimise the BIC score over the space of graphs. Furthermore, in the current revision, we added results of FLOP on non-linear, semi-synthetic, and real-world data in Appendix C.7, C.8 and C.9; in all these settings, FLOP search is efficient and effective for finding a graph with close-to-optimal BIC score. A key insight of our work is, that discrete search is a feasible and effective approach to finding graphs that optimize the score.
>
> - __Number of swap in ILS perturbations.__
> ILS restarts can never yield worse results with regard to BIC optimization; this is per definition as the graph obtained by a restart is only accepted if it yields an overall score improvement. We also, in almost all plots, show results of FLOP without ILS (i.e., when setting restarts to zero). To further corroborate our statement about $\ln p$ being a robust choice, we ran our algorithm for various other choices of $x \cdot \ln p$ and report the results in appendix C.10. Here, all choices work reasonably well in reducing SHD compared to running FLOP without ILS, with $x$ between $0.75$ and $2.0$ yielding almost identical results.
>
>
> __Questions__
>
> The reviewer did not include specific questions. We are, of course, happy to answer any questions if they arise during the discussion period or upon reading the other reviews and our responses.

---

> > ### Comment · Reviewer_UFtS · 2025-11-26
> >
> > Thank you for your response. I acknowledge that the author's idea is interesting. However, I regret to say that, considering
> > 1. the primary motivation of this paper is that finite-sample performance poses a practical challenge, and
> > 2. this paper investigates the most idealized case where the structural functions are linear, exogenous noises are Gaussian, and there is neither latent confounder nor selection bias,
> >
> > I still believe it is necessary to provide theoretical guarantees for finite-sample performance. Without this component, I feel that the completeness of this work falls considerably short of the standards expected for ICLR. Therefore, I have decided to maintain my original score.

---

> > > ### Author Response · Authors · 2025-11-26
> > >
> > > Thank you for acknowledging that the core idea of the paper is interesting.
> > >
> > > We respectfully but strongly disagree with the assertion that our work “falls short of ICLR standards” merely because it does not also provide finite-sample guarantees. Our contribution is technically sound and establishes new state-of-the-art performance in both accuracy and speed for score-based causal discovery.
> > >
> > > The linear Gaussian setting is theoretically well-understood in the asymptotic regime, yet remains the most studied base case in the causal discovery literature. The fact that the community continues to publish numerous influential papers on this setting without finite-sample guarantees underscores that the problem is far from “solved.” For example:
> > >
> > > - DAGs with NO TEARS: Continuous Optimization for Structure Learning, NeurIPS 2018
> > > - On the Role of Sparsity and DAG Constraints for Learning Linear DAGs, NeurIPS, 2020
> > > - DAGMA: Learning DAGs via M-matrices and a Log-Determinant Acyclicity Characterization, NeurIPS, 2022
> > > - Fast Scalable and Accurate Discovery of DAGs Using the Best Order Score Search and Grow-Shrink Trees, NeurIPS 2023
> > > - Extremely Greedy Equivalence Search, UAI 2024
> > > - QWO: Speeding Up Permutation-Based Causal Discovery in LiGAMs, NeurIPS 2024
> > >
> > > None of these works provide finite-sample guarantees, yet they are widely recognized as significant contributions. Our paper advances this line of research by demonstrating that discrete search (often dismissed as intractable) can be made highly efficient and accurate through principled algorithmic design, which was our primary motivation. This is a substantial and timely contribution that aligns well with ICLR’s standards.
> > >
> > > We hope this clarifies why we believe the reviewer’s expectation is misplaced and why our contribution stands on solid ground.

---

> > > > ### Comment · Reviewer_UFtS · 2025-11-26
> > > >
> > > > I respectfully disagree with the authors' claim. It is definitely acceptable for [1,2] to provide no theoretical analysis on finite-sample performance because their focus is not on how to improve the finite-sample performance. Instead, [1] aims to formulate the structure learning problem as a purely continuous optimization problem, while [2] aims to speed up the causal discovery algorithm. However, you repeatedly emphasize the importance of finite-sample performance in the Introduction.
> > > >
> > > > [1] DAGs with NO TEARS: Continuous Optimization for Structure Learning, NeurIPS 2018\
> > > > [2] QWO: Speeding Up Permutation-Based Causal Discovery in LiGAMs, NeurIPS 2024

---

### Official Review · Reviewer_ZiZP · 2025-10-26

**Soundness:** 3
**Presentation:** 3
**Contribution:** 4
**Rating:** 6
**Confidence:** 3

**Summary:**

This paper discusses the acceleration of score-based causal discovery algorithms. Traditionally, estimating causal DAGs from observational data using score-based criteria (e.g., BIC) has been regarded as computationally intractable due to the NP-hardness of discrete search, raising concerns about scalability in large datasets. In recent years, continuous optimization methods such as NOTEARS and DAGMA have emerged as popular alternatives. However, these approaches suffer from challenges including non-convex optimization, approximate enforcement of acyclicity constraints, and the need for post-hoc thresholding.

The paper proposes a new score-based causal structure learning algorithm, FLOP (Fast Learning of Order and Parents), which dramatically improves the efficiency of discrete search and achieves better performance than existing continuous optimization approaches. FLOP targets causal DAG estimation under a linear additive noise model (linear ANM). It introduces parent set reuse, where the parent sets of each node are initialized from the results of previous searches, reducing both computational and memory costs. A modified grow-shrink procedure is proposed and theoretically proven to correctly identify the Markov boundary. In addition, dynamic Cholesky updates enable efficient computation of conditional variances in the BIC score by reusing Cholesky decompositions, reducing the cost from $O(k^3)$ to $O(k^2)$. The method also incorporates principled order initialization, which constructs an initial variable order by placing highly correlated nodes adjacent to each other, and Iterated Local Search (ILS) to substantially improve the accuracy of discrete optimization.

Experimental results on synthetic data demonstrate that FLOP runs more than 100 times faster than conventional discrete search algorithms while achieving nearly the same accuracy as exact search methods. Moreover, it outperforms continuous optimization methods, highlighting the renewed importance of discrete optimization in score-based causal discovery.

**Strengths:**

While discrete search–based score methods have often been overshadowed by the recent rise of continuous optimization approaches due to their NP-hardness, this paper makes a significant contribution by demonstrating that discrete search can be made dramatically more efficient through theoretically grounded and well-designed algorithmic improvements. The proposed method introduces principled modifications to key components of the discrete search process, resulting in substantial computational gains.

Through simulation experiments, the authors show that their method runs over 100 times faster than conventional discrete search algorithms, achieving nearly the same accuracy as exact search methods. Moreover, it also outperforms representative continuous optimization methods. Overall, this work makes an important contribution to the field by reestablishing the relevance and potential of discrete optimization in score-based causal discovery, a domain recently dominated by continuous optimization approaches.

**Weaknesses:**

While this paper makes a valuable contribution by demonstrating that score-based causal discovery with discrete search can be made computationally practical through well-engineered algorithmic improvements, its theoretical and methodological novelty is somewhat limited. Many of the proposed components, such as parent set reuse, dynamic Cholesky updates, and local search, are not fundamentally new, but rather effective adaptations of existing techniques.

Moreover, although each of the algorithmic enhancements is well-motivated and complementary, their effectiveness may depend on specific data conditions. For example, the parent set reuse and dynamic Cholesky updates assume relatively stable local structures and well-conditioned covariance matrices; the principled order initialization relies on correlations being informative about causal directions; and the iterated local search may require tuning depending on the ruggedness of the score landscape. A more explicit discussion of these limitations and potential failure cases would further strengthen the paper’s credibility and generality.

**Questions:**

Could the authors discuss situations in which each of the proposed techniques, namely, parent set reuse, dynamic Cholesky updates, principled order initialization, and iterated local search, might fail to perform as expected?

---

> ### Author Response · Authors · 2025-11-20
>
> Thank you for the thoughtful review. We particularly appreciate your take that our results reestablish "the relevance and potential of discrete optimization in score-based causal discovery". You also raised two valuable concerns which we adress below.
>
> 1. __Novelty__
>
> It is true that dynamic Cholesky updates and iterated local search are established techniques in other areas of research. However, the application of ILS to order-based causal discovery is novel and the same holds, to the best of our knowledge, for updating the score in discrete search instead of recomputing it, which has not been exploited in causal discovery previously. The same holds for the parent set reuse in grow-shrink where all previous work started from the empty set.
>
> For pointers to related work, see also the answer to Point 3 in our reply to Reviewer cdXL.
>
> 2. __Failure cases__
>
> We agree that it is important to study failure cases for our algorithm, and the reviewer raises several scenarios:
>
> - _Cases where the covariance matrix is ill-conditioned._
> We rely on Cholesky decompositions, which are generally numerically stable. Our implementation also includes explicit checks for numerical instability in the update steps and the Cholesky decomposition can be recomputed from scratch whenever an update fails. For ill-conditioned covariance matrices, all algorithms based on the Gaussian BIC score face the same numerical dilemma; by calculating the score via the Cholesky decomposition FLOP relies on one of the numerically best-behaved ways to calculate the score (as opposed to, for example, calculating the residuals via instable matrix inversions).
>
> - _Settings where our starting order heuristic fails._
> Our starting-order heuristic does not assume that (conditional) correlations encode causal direction, but only that they capture some notion of proximity. After all, we are merely aiming for a starting order that avoids the issues of using a random order on path instances (where the candidate parents may be extremely weakly correlated). The search for the best order is then left to the iterated local search, which would just need many more restarts when started from a random order.
>
> - _Settings with particularly rugged search spaces._
> Iterated local search can never harm the optimization performance of a discrete search algorithm. In our simulation study, FLOP was almost always able, given a sufficiently large number of ILS runs, to find or come close to a BIC-optimal graph. This is true even in settings where the true graph does not minimize BIC (see Appendix C.4), which indicates that our ILS heuristic can avoid being trapped in even quite rugged search landscapes. There may be cases that require a very large number of restarts (e.g., the dense graph setting discussed below).
>
> - _Settings where the local structures are unstable._
> If parent sets (needed to) change a lot upon reinserting a node (and the local structures were "unstable" in this sense), then this would only lead to slower run-times (as we would not benefit from the speedups that parent set reuse oftentimes provides), but it would not prohibit the search from finding well-scoring graphs. In fact, as we show in Lemma 3.2, the grow-shrink routine is correct when started from any previous parent set.
>
> In the paper, we report two distinct failure cases:
> - Cases where the BIC optimization of FLOP is suboptimal, such as the dense ER graphs in Figure 8 (appendix C.4). Here, an exact score-based algorithm yields clearly better results even compared to FLOP with 500 ILS iterations. This shows that BIC optimization is the problem here and that even more ILS restarts would be needed. Very dense graphs are some of the hardest instances with regard to BIC optimization for FLOP that we encountered (dense graphs are also considered hard instances for most other score-based algorithms).
> - Cases where graphs close to the BIC optimum do not yield good SHD results. These issues are caused by the scoring criterion and not the search, which we focus on in this paper. We give the Pathfinder bnlearn instance as an example (Figure 11), for which BIC improvements even lead to worse SHD. Moreover, we added results for non-linear, semi-synthetic and real world-data (Appendix C.7, C.8 and C.9) where we observe that despite excellent BIC optimization FLOP may find graphs with a large distance to the ground-truth.

---

### Official Review · Reviewer_cdXL · 2025-10-26

**Soundness:** 2
**Presentation:** 4
**Contribution:** 3
**Rating:** 6
**Confidence:** 3

**Summary:**

Summary. The authors propose a modification of the BOSS algorithm (Andrews et al., 2023), a score-based approach for Markov Equivalence Class discovery from data generated according to a linear Gaussian model. The authors propose 4 modifications to improve the efficiency and accuracy of the algorithm. They show that their modifications preserve theoretical guarantees of convergence of the method to a solution in the Markov equivalence class. The empirical results support the claim of improved scalability and accuracy.

**Strengths:**

The main contribution of this paper is an algorithm with really impressive empirical performance, both in terms of accuracy and scalability.  Theoretical guarantees of identifiability of the Markov equivalence class that are common in score-based approaches are maintained, as the authors demonstrate. Finally, the empirical results are convincingly and well presented. Figure 2 clearly shows improvements in the run time. Figure 1,3,4 clearly show the improvements in terms of accuracy due to the ingredients they add to their algorithm. Nice work!

**Weaknesses:**

1. **Real world  / beyond linear Gaussian experiments.** The proposed method achieves impressive performance on linear Gaussian data, which are covered by the theory of the algorithm. However, the scope of these models is quite limited. It would be relevant to experiment beyond linear Gaussian synthetic data (at least nonlinear ANM + vary the noise distribution — analytic or NN generated nonlinearities, e.g., see [Montagna et al. 2023](https://causally.readthedocs.io/en/latest/generated/causally.scm.causal_mechanism.NeuralNetMechanism.html#causally.scm.causal_mechanism.NeuralNetMechanism)), and also with real-world datasets (that’s where causal discovery is interesting at the end of the day). E.g. you can consider the following datasets that are easy to find: bnlearn networks considered in the paper (please use real data when available), Sachs, Lucas, Dream (relevant for high dimensions), and the Syntren synthetic data generator.
2. **Hyperparameter tuning**. The authors set $\lambda_{BIC} = 2$. It is, however, unclear whether this is optimized for the analysed datasets, and how sensitive FLOP is to this hyperparameter. Could the authors discuss that, and present experimental results at different values of $\lambda_{BIC}$?
3. **Incremental nature of the contribution**. It is worth noting that the new method consists of smart but incremental additions to existing methodologies — this doesn’t change my opinion on the positive value of the work. In this sense, I am curious about whether the *warm start* tweak of Section 3.1 and the use of ILS in Section 4.2 are novel in causal discovery, or whether the authors have a reference to past use of these ideas.

I am happy to raise my score upon satisfactory execution of the required experiments!

**Questions:**

What are all the hyperparameters of the method? In the weaknesses I ask clarifications about $\lambda_{BIC}$, but are there others? If yes, can the authors clarify how the hyperparameters are set, and empirically analyse the sensitivity of FLOP to changes?

---

> ### Author Response · Authors · 2025-11-20
>
> Thank you for the constructive review. Below, we address your comments and questions point-by-point.
>
> 1. __Real world / beyond linear-Gaussian experiments__
>
> The focus of the paper was to showcase what performance is possible in the standard problem class (linear Gaussian) when fully embracing discrete search. As a result, we focused on this (synthetic) setting in our simulations. However, we agree that it is important to consider examples outside this setting. We have added results for non-linear ANMs sampled from random multi-layer perceptrons with one hidden layer (the setting from the DAGMA paper, appendix C.2.2) as well as randomly sampled Gaussian processes with a unit bandwidth RBF kernel (the setting from Rolland et al. (2022), "Score Matching Enables Causal Discovery of Nonlinear Additive Noise Models") and report the results in Appendix C.7. It can be seen that FLOP still optimizes the BIC well, however, as the linear Gaussian BIC is misspecified in this setting, score-optimal graphs do not recover the ground-truth here. Furthermore, we added results on the semi-synthetic causalAssembly data in Appendix C.8 where FLOP achieves state-of-the-art in graph recovery, even though the SHD is quite high. Here, we observe that the BIC scores are significantly better than the BIC of the true DAG, indicating that the optimization is not the obstacle for further improvements (see statistical vs search problem in the general response). Finally, in Appendix C.9, we report results on the real-world Sachs data set, where FLOP obtains the score-optimal graph (also found by BOSS and GES).
>
> 2. __Hyperparameter tuning__
>
> We deliberately chose not to tune the penalty parameter in the BIC, instead opting for a standard community choice: $\lambda=2$ (this is, for example, the default in XGES mentioned above and many other works such as GRaSP and BOSS). There is also theoretical evidence from high-dimensional consistency results suggesting that penalty values larger than $1$ have the best small sample characteristics ("Extended Bayesian Information Criteria for Gaussian Graphical Models" by Foygel, Drton, NeurIPS 2010). We also note that asymptotic consistency holds for all constant choices of  $\lambda$. However, we agree that additional insight into the role of the penalty is helpful, so we have added a simulation study with different BIC penalties in Appendix C.10. The results are in line with the theoretical works mentioned above.
>
> 3. __Incremental nature of the contribution__
>
> Yes, we are happy to give more background on related work here and added further references in the paper:
> - Efficient Cholesky updates are a textbook technique, albeit to the best of our knowledge this is the first work to avoid recomputation of the local score in discrete search for Bayesian network structure learning/causal discovery, opting for efficient updates instead.
> - Iterated local search is a standard technique for utilizing compute to improve optimization results. In the context of Bayesian network structure learning, it is a rarely used technique. It has been applied by Liu, Gao, Wang, Ru and Zhang in "A metaheuristic causal discovery method in directed acyclic graphs space" (2023) together with ideas from genetic algorithms (the code is unfortunately not available). The XGES algorithm mentioned by reviewer MbRD can also be seen as a version of ILS. We reference both in the new revision of the paper. FLOP is the first method to use ILS in order-based causal discovery. Moreover, the effectiveness of ILS in FLOP stems from its run-time improvements, which allows for effectively leveraging compute (compared to e.g. XGES which is slower and overall less effective, see point 1 in our response to the questions of reviewer MbRD).
> - The idea of starting the grow-shrink with the previous parent set is, to the best of our knowledge, novel in the context of causal discovery. We are not aware of other works that use this technique.
> - There are causal discovery algorithms that rely on constructing an (initial) order, subsequently learning the parents and immediately returning the found DAG without further search (such as DirectLiNGAM). However, previous order-based search algorithms such as GRASP and BOSS generally start with a random order. We are also the first to address the issue of these algorithms on path instances, where initalizing with a random order harms performance.

---

> > ### Author Response · Authors · 2025-11-20
> >
> > 4. __Other hyperparameters__
> >
> > Thanks for the question! The central hyperparameter of FLOP is the BIC penalty parameter (see the answer to point 2 for the sensitivity of FLOP to this parameter choice).
> >
> > Another parameter that can be set is the number of ILS iterations. This is not a hyperparameter in the traditional sense as, by definition, more ILS iterations can only improve the BIC optimization. Hence, this parameter trades off increased run-time with better optimization. We include multiple choices for the number of ILS iterations in most plots in the paper.
> >
> > Further, there is an internal parameter in our software, namely the number of random swaps that are performed in a perturbation of the order, which amounts to $\log p$ by default. We did not make this parameter tuneable, but we still offer an overview of its effect in appendix C.10 (see also our answer to weakness 3 of reviewer UFtS).

---

> > ### Comment · Reviewer_cdXL · 2025-11-23
> >
> > 1. Thanks for the additional experiments on real-world and synthetic nonlinear data. They are important to make the contribution more transparent. In this regard
> >     1. In Appendix C.7, the authors do not mention how the synthetic graphs are generated, the number of nodes, or the number of edges in the synthetic graphs: without this information, it is quite hard to evaluate whether an SHD is bad or good. I would also add the information on the number of edges in the caption of the figure.
> >     2. For the experiments on Causal Assembly and Sachs, I have the same issue: it is hard to understand what a *good* SHD value corresponds to, without knowing the number of edges.
> >
> >     Given the poor performance results in these settings, it would be better to include something like a limitation box in the main text pointing to the appendix that the method has poor performance beyond the assumptions. Similarly, I would clearly remark in the appendix section, where all the experiments are run, that FLOP doesn’t look reliable beyond the assumptions.
> >
> > 2. The experiments on $\lambda_{\textnormal{BIC}}$ different values are satisfying. Thanks!
> > 3. Thank you for the discussion about the past use of the ideas in the paper and for adding the right citations.

---

> > > ### Author Response · Authors · 2025-11-24
> > >
> > > Thank you for your quick response and for catching the missing details in the new Appendices C.7–C.9. We have added that information to the revised PDF, including figure captions, similar to the level of detail provided for the original experiments. This was an oversight as we prioritized getting the new experiments to you quickly.
> > >
> > > Regarding performance beyond the linear Gaussian setting: FLOP does what it is designed for, which is optimizing the Gaussian BIC score. When the score is well-specified, this translates into excellent recovery. When it is misspecified, good recovery cannot be expected. This highlights that the bottleneck in these regimes is the scoring criterion, not the optimization (which FLOP solves well). We see this as a strength because FLOP makes this distinction transparent and shows that discrete search is no longer the main obstacle (oftentimes achieving SHD of 0 when the assumptions are met and the sample size is sufficient for the asymptotics known for the Gaussian BIC score to kick in). To make this clearer, we have expanded the discussion section to refer to the new simulation results in the appendix and frame this as an agenda for future work: developing suitable scores for broader settings now that we know discrete optimization can be feasible. As suggested, we also highlight these results in the opening paragraph of Appendix C.
> > >
> > > Thank you again for helping us improve clarity.

---

> > > > ### Comment · Reviewer_cdXL · 2025-11-27
> > > >
> > > > Thank you for considering my concerns and implementing them in the new version of the paper. I am now happy to recommend acceptance, and I raised my score to 8.

---

### Official Review · Reviewer_MbRD · 2025-10-28

**Soundness:** 4
**Presentation:** 3
**Contribution:** 4
**Rating:** 6
**Confidence:** 4

**Summary:**

This paper introduces FLOP (Fast Learning of Order and Parents), a highly efficient score-based algorithm for causal structure learning in linear models. FLOP significantly accelerates the reinsertion-based local search over topological orders by "warm-starting" the grow-shrink parent selection and using $O(k^2)$ Cholesky rank-one updates for score computation, avoiding $O(k^3)$ re-computation . This speed advantage is leveraged to run an Iterated Local Search (ILS) with a principled order initialization, enabling the algorithm to escape local optima and achieve accurate graph recovery with dramatically lower runtime. The work compellingly challenges the conventional notion that discrete search cannot scale outside of small graphs.

**Strengths:**

This paper builds upon a prior discrete search algorithm BOSS, by proposing 4 distinct modifications: 1) reusing parent sets for the grow-shrink procedure to improve runtime, 2) dynamically updating the choleksy decomposition to improve runtime, 3) initializing the first order to mitigate against known failure modes of order-based methods, and 4) running iterated local search to escape local minima.

Modifications 1), 2), and 3) seem entirely original to my knowledge (for 4) see the authors response to my first question in the Questions section). The quality of the work is high, as proofs provided make sense, all notation is clear, standard definitions and concepts in causal discovery are consistently used, and extensive coverage of the relevant literature is explored. The paper is generally clear, with a few caveats that I discuss in the Weaknesses and Questions section. Most importantly, the empirical results concerning the reduction in runtime are quite significant - in particular, Figure 2 shows how the addition of each modification leads to substantial decreases in the runtime needed for a fixed number of nodes, enabling scaling to graphs with over 500 variables. This is extremely significant given the fact that most discovery algorithms in the literature fail to scale past 30-50 nodes. In general, FLOP builds upon a growing trend of papers [1,2, 3] exploiting local structure previously ignored by methods aiming only for consistency. Additionally, this paper puts forward ideas that could be very helpful for other researchers in the field, even beyond score-based subfield.

References:

[1] Hybrid Top-Down Global Causal Discovery with Local Search for Linear and Nonlinear Additive Noise Models., Hiremath et al., NeurIPS 2024

[2] LoSAM: Local Search in Additive Noise Models with Mixed Mechanisms and General Noise for Global Causal Discovery., Hiremath et al. UAI 2025

[3] Strong and Weak Identifiability of Optimization-based Causal Discovery in Non-linear Additive Noise Models, Li et al., ICML 2025.

**Weaknesses:**

1. The empirical analysis is conducted largely on simulated linear settings, with only a few real-world dataset considered. Given that the main contribution of FLOP is strictly better empirical performance (no theoretical guarantees outlining provable improvements are given), a more substantial empirical study of the performance of FLOP in nonlinear ANM and real-world data would help contextualize the contribution of FLOP. This is true even if results are not as promising as in the linear setting, as it might highlight where modifications in future could be made to build on top of FLOP.

 2. Although the authors differentiate FLOP from BOSS at some specific subparts of the algorithm, (for ex. at the end of Section 2 they highlight the BOSS reinsertion strategy to the reader to build upon), it is difficult for a reader who is not already familiar with the BOSS paper to parse. For example, at the beginning it’s not clear whether Algorithm 1 constitutes the entirety of the BOSS algorithm, or just a subpart (it is, which is discussed later). Would be nice for the full algorithm environments of BOSS and FLOP and corresponding walk-throughs to be present (or if necessary signaled to the reader early on and placed in the Appendix) to facilitate easy comparison for the reader. Or perhaps even adding a figure describing the general pipeline of these specific score-based methods could be helpful, for better clarity.

**Questions:**

1. How does this approach compare to heuristic adaptions of GES such as XGES [1], where heuristics similar to ILS are used to escape local minima? Theoretical and experimental comparison would both be useful here.

2. How does the proposed approach here compare empirically against recent score-based methods such as GRaSP [2], or against FCM methods designed for linear ANMs, such as DirectLiNGAM [3]? In general, it would be useful for the authors to undertake a more comprehensive empirical comparison to other recent methods, as many of the paper’s contributions relate to the promise of improved empirical performance, but without any formal guarantees of improvement.

3. To what extent can the heuristic modifications made in FLOP be translated to the setting of discrete search in nonlinear ANMs? Some modifications seem like they might likely generalize such as the modified grow-shrink procedure, but other improvements such as dynamic Cholesky updates seem only possible due to the linearity assumptions.

4. Is it possible to recalculate the ordering metric to use some more common notion of topological divergence, as seen unnormalized in [4] or normalized in [5,6]? This would provide additional insight into the performance of FLOP.

5. Please give more details on this: “The standard hardness constructions rely on data-generating processes that cannot be represented by a DAG over the observed variables (Chickering, 1996; Chickering et al.,2004).” Not sure if I understand correctly what it means for a DGP to not be representable as a DAG over the observed variables (do you mean they require unobserved variables?).

6. In Figure 2, can the authors clarify what the difference between FLOP naive and FLOP (reusing parents) is? My understanding is that the algorithm generated by modifying grow-shrink is exactly FLOP? If not, what exactly is the base FLOP algorithm?

References:

[1] Extremely Greedy Equivalence Search, Nazaret et al., UAI 2025.

[2] Greedy Relaxations of the Sparsest Permutation Algorithm, Lam et al., UAI 2022.

[3]DirectLiNGAM: A Direct Method for Learning a Linear Non-Gaussian Structural Equation Model, Shimizu et al., JMLR 2011.

[4]Causal Discovery with Score Matching on Additive Models with Arbitrary Noise, Montagna et al., CleaR 2023.

[5] Hybrid Top-Down Global Causal Discovery with Local Search for Linear and Nonlinear Additive Noise Models., Hiremath et al., NeurIPS 2024

[6] LoSAM: Local Search in Additive Noise Models with Mixed Mechanisms and General Noise for Global Causal Discovery., Hiremath et al. UAI 2025

---

> ### Author Response · Authors · 2025-11-20
>
> Thank you for your thoughtful and detailed review. We appreciate the time and effort you put into providing construtive feedback. Below, we address your comments point-by-point and respond to each of your numbered questions.
>
> __Re Empirical Analysis & Algorithm Descriptions__
>
> 1. In Appendix C.7, C.8 and C.9 of the current revision, we added experimental results for non-linear, semi-synthetic, and real-world data, and discuss the strengths/weaknesses and further potential of FLOP in these settings.
> 2. Thank you for this excellent suggestion. In Appendix A, we added a high-level visualization of FLOP together with a summary of its differences compared to BOSS.
>
> __Re Questions__
>
> 1. Indeed, the XGES heuristic of removing a single edge that temporarily worsens the score combined with a subsequent local search is a version of an iterated local search (ILS). We have added a reference in Section 4.2 of the new revision. Empirically, we ran XGES on the standard setting in the paper with ER graphs containing 50 nodes and having average degree 8. Here, XGES achieves a mean SHD of 188.62 with average run-time of 141 seconds. This is clearly better than what standard GES achieves (SHD of around 300) but far from the performance of BOSS or FLOP. The main downside of XGES is that the GES operators are (i) computationally expensive and (ii) more local than the node-order-reinsertions in BOSS or FLOP and thus the algorithm is more prone to being stuck in local optima, which the ILS based on perturbing by a single-edge removal helps to escape only to a limited extent.
> 2. We also ran GRaSP on our standard setting (ER, 50 nodes, average degree 8) and it achieves, on average, an SHD of 13.34 and a run-time of 7.73 seconds. From our experience BOSS typically gives at least as good of a performance as GRaSP while being slightly faster (as also shown in the BOSS paper). Moreover, we observed that GRaSP shares the issues with BOSS on the path instances (cf. Section 4.1). We also ran DirectLiNGAM on the ER graph setting where it only achieves an average SHD of 602.76 due to the noise being Gaussian. Additionally, we ran DirectLiNGAM for the newly added experiments on non-linear, semi-synthetic data, and real-world data, and report the results in Appendix C.7, C.8 and C.9.
> 3. Great question! Yes, the modified grow-shrink procedure can directly be transfered to this setting. The same holds for ILS. While the Cholesky updates themselves are tailored to the linear case and the Gaussian BIC score, it is conceivable that other scoring criteria can in a similar spirit be iteratively updated instead of recomputed. We agree that this is an interesting avenue for further research.
> 4. We are unsure whether we understand this point correctly. We have not explored further alternative ordering criteria/"ordering metrics" for initialization. Our initialization (cf. Section 4.1) successfully addresses the failure mode we observed in other order-based algorithms that start from a random order (which would require a lot more ILS restarts to obtain comparable results) and with it FLOP achieves competitive graph recovery results.
> 5. Yes, Chickering's proofs rely on unobserved variables. We updated the sentence in the introduction to clarify: "The standard hardness constructions rely on data-generating processes that involve unobserved variables and cannot be represented by a DAG over only the observed variables (Chickering, 1996; Chickering et al., 2004)."
> 6. Thank you for this clarification request! We completely agree and have updated the labels in Figure 2 accordingly. The fastest displayed algorithm is indeed FLOP. The other two algorithms are ablations, where the run-time optimizations described in Section 3 are not used (we now term this "pre-FLOP" in the Figure), which we include to quantify the achieved run-time gains.

---

### Author Response · Authors · 2025-11-20
**General Response to the Reviewers**

We thank all reviewers for their thoughtful and constructive feedback. The comments have helped us identify ways to improve presentation and add context.

__Core Contribution and Clarifications__

It is great to see that all reviewers appreciate the central idea of the paper. For example, Reviewer MbRD writes that our work "compellingly challenges the conventional notion that discrete search cannot scale outside of small graphs" and Reviewer cdXL highlights that FLOP "brings together several smart incremental additions" to make discrete search competitive. Reviewer ZiZP notes that "Overall, this work makes an important contribution to the field by reestablishing the relevance and potential of discrete optimization in score-based causal discovery, a domain recently dominated by continuous optimization approaches".

At the same time, the reviews have helped us identify areas to clarify:

- __Our main contribution is on search, not proposing a new score (statistical vs search problem).__

In score-based causal discovery, there are two separate questions:

    1) Is the true graph score-optimal? (statistical)
    2) Can we find a score-optimal graph? (search)

FLOP tackles question 2) and in practice finds graphs with close-to-optimal BIC scores. This shows that, contrary to common claims, effective BIC score optimization through discrete search is possible. Once this is achieved, how well a BIC-optimal graph recovers the true graph depends solely on the scoring criterion, not the optimization algorithm. We have emphasized this distinction further by additionally reporting the BIC score in the ER and SF graph settings (Figure 5) as well as in the newly added experiments on non-linear, semi-synthetic, and real-world data (Appendix C.7, C.8 and C.9; see also below).

- __Synthetic data vs real-world data.__

In our initial submission, the simulations focused on synthetic data where the model assumptions are met and the ground-truth graph is known. On these settings, FLOP obtains state-of-the-art graph learning results, corroborating the claim that discrete search for well-scoring graphs is feasible and effective. In the current revision, we further include results on non-linear data (non-linearities generated by MLPs and by sampling Gaussian Processes), semi-synthetic data (causalAssembly), and real-world data (Sachs). Here, as we show, FLOP is still successful in optimizing the Gaussian BIC score, while the quality of the found well-scoring graphs then depends on whether the scoring criterion is suitable to recover the ground-truth graph. This also highlights opportunities for future work: to design suitable scoring criteria that can be evaluated and updated efficiently, enabling effective discrete search for causal discovery beyond the linear Gaussian setting.


__Updates to the Paper__

In the current revision, we added:
- Appendix C.7, C.8 and C.9 which show additional experimental results on non-linear data, the causalAssembly semi-synthetic data, and the Sachs real data set.
- Figures 5, 12, and 13 showing the BIC scores of the learned graphs in multiple settings, which we compare to the SHD results.
- Appendix A, which features a new visualization of FLOP's high-level control flow and delineates how FLOP differs from the BOSS algorithm.

---

### Meta-Review · Area_Chair_TaoL · 2026-01-06

**Summary:**

This paper introduces a highly efficient score-based algorithm for causal structure learning in linear Gaussian setting, by proposing four modifications to existing method to improve the efficiency and accuracy of the algorithm. The reviewers generally appreciated the methodological contributions and superior empirical performance, both in terms of accuracy and scalability. While there are a few initial concerns about e.g. the experiments, clarity of algorithm, and comparisons to related methods, most of them were addressed during the rebuttals, and I thereby recommend acceptance.

I would encourage the authors to take into account all reviewers' feedbacks in future version of the manuscript, including those further concerns by Reviewer UFtS (e.g., discussion of existing works that support the relevance of linear Gaussian setting, as well as why finite-sample theoretical analysis is of a difference scope). On a related note, I noticed one of the baselines is DAGMA, which relies on the variances of noises being equal. This may not be fair as it leads to model misspecification (the simulation settings do not adopt such an assumption), and thus I would encourage the authors to use other continuous-based linear-Gaussian baseline that does not rely on such an assumption.

**Reviewer Concerns:**

Concerns addressed by rebuttal:
- Limited evaluation beyond linear-Gaussian settings (Reviewers MbRD, cdXL): addressed via added nonlinear, semi-synthetic, and real-world experiments, with clear discussion of score misspecification.
-  Comparisons to related methods (Reviewer MbRD): addressed with new empirical results and clearer positioning.
- Algorithm clarity and relation to BOSS (Reviewer MbRD): addressed through high-level visualizations and discussion of differences compared to BOSS.
- Sensitivity analysis of hyperparameter (Reviewers cdXL): addressed through additional experiments and discussions.
- Incremental nature of the contribution (Reviewers cdXL, ZiZP): addressed through a detailed discussion of key improvements.
- The paper considers only the most idealized linear Gaussian setting (Reviewer UFtS): addressed through a detailed discussion of why such a setting is important.

Outstanding concerns:
- Lack of finite-sample theoretical guarantees (Reviewer UFtS): remains unresolved, but this reflects a difference in scope rather than a limitation, and is not viewed as critical by other reviewers.

**Reviewer Scores:**

Reviewer cdXL decided to increase their score to 8, although it is not reflected on the review. Reviewers MbRD and ZiZP may have increased their scores to 8 if they had been able to participate fully in the discussion.

---

### Decision · Program_Chairs · 2026-01-26

Accept (Poster)